# MODELING COMPLEX SYSTEM DYNAMICS WITH FLOW MATCHING ACROSS TIME AND CONDITIONS

**Martin Rohbeck**[1,2,3,4,*], **Edward De Brouwer**[1], **Charlotte Bunne**[1,5], **Jan-Christian Huetter**[1], **Anne Biton**[1], **Kelvin Y. Chen**[6], **Aviv Regev**[1,†], **Romain Lopez**[1,5,†]

[1]Genentech, USA
[2]Heidelberg University, Germany
[3]German Cancer Research Center, Germany
[4]European Molecular Biology Lab, Germany
[5]Stanford University, USA
[6]Osaka University, Japan

## ABSTRACT

Modeling the dynamics of complex real-world systems from temporal snapshot data is crucial for understanding phenomena such as gene regulation, climate change, and financial market fluctuations. Researchers have recently proposed a few methods based either on the Schrödinger Bridge or Flow Matching to tackle this problem, but these approaches remain limited in their ability to effectively combine data from multiple time points and different experimental settings. This integration is essential in real-world scenarios where observations from certain combinations of time points and experimental conditions are missing, either because of experimental costs or sensory failure, and ultimately to uncover general principles shared across different temporal processes. To address this challenge, we propose a novel method named *Multi-Marginal Flow Matching* (MMFM). MMFM first constructs a flow using smooth spline-based interpolation across time points and conditions and regresses it with a neural network using the classifier-free guided Flow Matching framework. This framework allows for the sharing of contextual information about the dynamics across multiple trajectories. We demonstrate the effectiveness of our method on both synthetic and real-world datasets, including a recent single-cell genomics data set with around a hundred chemical perturbations across time points. Our results show that MMFM significantly outperforms existing methods at imputing data at missing time points.

## 1 INTRODUCTION

Understanding the dynamics governing complex systems is essential for modeling their underlying mechanisms as well as for predicting future behavior, such as stock price fluctuations (Gontis et al., 2010), climate change (Franzke et al., 2015), and gene regulation leading to cell differentiation or response to drugs (Yeo et al., 2021; Schiebinger et al., 2019). In many real-world applications, measurements are collected at various time points and under diverse conditions, such as different environmental factors or experimental settings. Additionally, measurements are often unpaired across time and conditions. This may be attributed to the difficulty of matching patients across cohorts (Manton et al., 2008) in a clinical context or to the destructive nature of single-cell genomics (Ding et al., 2022) in modern experimental assays. This results in an extensive yet fragmented view of the system, necessitating models that can effectively integrate these data.

Such data fragmentation is prevalent in the field of drug development. For example, researchers aim to document the response of cells to various genetic and chemical perturbations in order to understand mechanisms of action and predict responses to novel conditions (Rood et al., 2024). Historically, technological and budgetary constraints limited researchers' ability to generate datasets that

---

\* This work was conducted during an internship at Genentech.
† Correspondence to: regeva@gene.com, romain.lopez@nyu.edu

captured cell profiles either across time for a single condition or across multiple conditions but at a single time point. Existing methods reflect these limitations, either modeling conditions without temporal components (Yu & Welch, 2022; Lotfollahi et al., 2023; Piran et al., 2024; Bunne et al., 2022; 2023) or modeling dynamics without interventions (Schulz et al., 2012; Tong et al., 2020; Wang et al., 2023). Recent advances in screening technologies enable cost-effective, large-scale measurement of cell populations over time under multiple conditions. This advancement motivates the development of new modeling strategies that can effectively leverage this combined complexity. Moreover, even with such advances, the sheer scale of biology and chemical space makes it prohibitive to experimentally characterize the entire temporal and perturbation space. When human patient data are concerned, these abilities are further diminished. Thus, better models are essential to generalize to unseen lab experiments and into human biology.

To leverage the rich and complex data now available and deliver better models, we propose framing the problem of learning population dynamics as modeling the transport of the probability distribution of cellular states across time and conditions. This approach allows us to capture changes over time and across conditions, while accommodating unpaired samples. Recent advances in generative modeling, such as diffusion models (Ho et al., 2020) and Flow Matching (FM) (Lipman et al., 2023), constitute effective methods for learning mappings between arbitrary distributions. This makes them particularly suitable for our task. However, despite the popularity of these approaches, they have been hitherto limited to transporting between two marginal distributions. Although multiple time points could be naively modeled by decomposing the problem into a series of two-marginal problems, such an approach would have significant limitations. It would fail to (1) leverage dependencies between the dynamics of multiple (more than two) related conditions (*i.e.,* similar conditions are assumed to follow similar dynamics), and (2) incorporate prior knowledge or constraints on the long-range dynamics of the system.

To address the aforementioned challenges, we introduce Multi-Marginal Flow Matching (MMFM), a general Flow Matching framework that learns system dynamics from populations measured at multiple time points and under various conditions. Notably, our model uniquely leverages dynamic dependencies between conditions and enables both interpolation and extrapolation to unobserved time points and/or conditions. Through extensive experiments, we demonstrate MMFM's superior performance and generality compared to existing methods. We apply our MMFM to a real-world dataset of immune cells subjected to each of a hundred kinase inhibitor perturbations measured over four time points and to a dataset of air pollutants concentrations. Empirically, our method successfully predicts cellular trajectories and air pollutants concentrations for unobserved conditions and time points.

## 2 BACKGROUND

In this section, we introduce the concepts of vector fields and probability density paths and then provide an overview of the Flow Matching (FM) framework. Unless mentioned otherwise, $\mathcal{X}$ denotes an Euclidean space (i.e., $\mathcal{X} = \mathbb{R}^d$).

**Vector Fields**  We model the underlying dynamics of a system using a time-dependent[1] vector field $u : [0, 1] \times \mathcal{X} \to \mathcal{X}$. This vector field induces a time-dependent diffeomorphic map on $\mathcal{X}$, called a flow, $\phi : [0, 1] \times \mathcal{X} \to \mathcal{X}$, defined by the Ordinary Differential Equation (ODE)

$$\frac{d}{dt}\phi_t(x) = u_t(\phi_t(x)), \tag{1}$$

with the initial condition $\phi_0(x) = x$. Given an initial probability distribution $p_0 : \mathcal{X} \to \mathbb{R}^+$, the flow reshapes the distribution $p_0$ into a *probability density path* $p_t(x)$ defined via the push-forward equation $p_t(x) = [\phi_t]_* p_0(x)$. The resulting density for each time $t$ is then obtained via the change of variable formula.

**Flow Matching**  The original FM framework for generative modeling (Lipman et al., 2023) considers two marginal distributions: the source distribution $p_0$ (e.g., Gaussian random noise) and the target distribution $p_1$ (e.g., a set of images), both with support contained in $\mathcal{X}$. The goal is to approximate the target vector field $u_t$ that generates a probability path $p_t(x)$ satisfying the boundary

---

[1]We use the subscript notation for the time parameter, e.g., $u_t(x)$.

conditions $p_0$ and $p_1$ with a trainable vector field $v_t(.\,;\theta)$ parameterized by $\theta$. This leads to the Flow Matching objective:

$$\mathcal{L}_{\text{FM}}(\theta) = \mathbb{E}_{t,x\sim p_t(x)} \|v_t(x;\theta) - u_t(x)\|_2^2, \tag{2}$$

with $t$ sampled from the uniform distribution $t \sim \mathcal{U}([0,1])$. Given an approximation of the target vector field $u_t(x)$, one can easily sample from the distribution $p_1$ given samples from $p_0$ (or the other way around), as well as predict dynamics of individual samples over time. However, the objective function in Equation 2 is generally intractable, as both $p_t$ and $u_t$ are unknown. Lipman et al. therefore introduced Conditional Flow Matching (CFM), a tractable yet equivalent objective that aims to approximate a *conditional* vector field $u_t(x \mid z)$:

$$\mathcal{L}_{\text{CFM}}(\theta) = \mathbb{E}_{t,z\sim q(z),x\sim p_t(x|z)}\left[\|v_t(x;\theta) - u_t(x \mid z)\|_2^2\right]. \tag{3}$$

The conditioning variable $z$ and conditional probability paths $p_t(x \mid z)$ are chosen such that the marginals match the boundary distributions $p_0$ and $p_1$. In general, one chooses $z$ as a pair of samples from the source and target distributions $(x_0, x_1)$ according to a joint distribution $q(z) = \pi(x_0, x_1)$ with marginals $p_0$ and $p_1$. Remarkably, $\mathcal{L}_{\text{FM}}(\theta)$ and $\mathcal{L}_{\text{CFM}}(\theta)$ are equivalent objectives as they have identical gradients with respect to $\theta$ (Theorem 2 of Lipman et al. (2023)).

**Conditional Probability Paths**   A common choice for the form of conditional probability paths is

$$p_t(x \mid z) = \mathcal{N}(x \mid \mu_t(z), \sigma_t(z)^2 I). \tag{4}$$

where $\mu : [0,1] \times \mathcal{Z} \to \mathcal{X}$ and $\sigma : [0,1] \times \mathcal{Z} \to \mathbb{R}^+$ represent the time-dependent mean and standard deviation of a Gaussian distribution. Since the number of potential vector fields inducing a desired path is infinite, a reasonable approach is to select a *simple* vector field, *e.g.*, one that results in a flow of the form:

$$\phi_t(x \mid z) = \mu_t(z) + \sigma_t(z)\left(\frac{x - \mu_0(z)}{\sigma_0(z)}\right). \tag{5}$$

The above functional form leads to a unique inducing vector field (Theorem 3 in Lipman et al. (2023)), defined as

$$u_t(x \mid z) = \frac{\sigma_t'(z)}{\sigma_t(z)}(x_t - \mu_t(z)) + \mu_t'(z). \tag{6}$$

Given this parameterization, different $\mu_t(z)$ and $\sigma_t(z)$ are possible, as long as the marginals coincide with the boundary conditions (*i.e.*, $\int p_0(x \mid z)q(z)dz \approx p_0(x)$ and $\int p_1(x \mid z)q(z)dz \approx p_1(x)$). The most natural is to consider a small variance function at the boundaries with an interpolation function for $\mu_t(z)$, such that $\mu_0(z) = x_0$ and $\mu_1(z) = x_1$. For instance, Tong et al. (2024) considered a linear interpolation for the mean $\mu_t(z) = tx_0 + (1-t)x_1$, with a constant (small) variance $\sigma_t(z) = \sigma > 0$, and $q(z) = p_0(x_0)p_1(x_1)$, for their Independent-CFM (I-CFM) method.

**Optimal Transport**   Tong et al. (2024) highlighted that sampling points $x_0$ and $x_1$ independently (*i.e.*, $\pi(x_0, x_1) = p_0(x_0)p_1(x_1)$) could lead to training instabilities and poor performance. Their improved method, OT-CFM, instead sampled using the optimal transport coupling between $p_0$ and $p_1$, $\pi^*(x_0, x_1)$, and resulted in stabilized training and better performance.

## 3   Condition-aware Multi-Marginal Flow Matching

We first extend the Flow Matching framework to multiple temporal snapshots, and then present a condition-aware variant that leverages information across various marginals and contexts.

**Problem Statement**   Generalizing the Flow Matching framework that considers two distributions $p_0$ and $p_1$ only, we assume the data is sampled from a collection of distributions $\{p_{t_k}(x \mid c) : c \in [C], k \in [K+1]\}$ over $C$ different conditions and $K+1$ different sampling times. Given these observations, our goal is to learn an approximation of a target vector field $u_t(x \mid c)$ that generates a probability path $p_t(x \mid c)$ that satisfies the boundary conditions at $t_0, t_1, \ldots, t_K$ for all conditions.

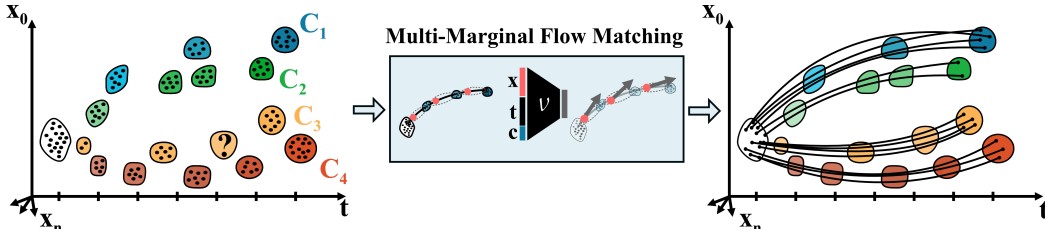

Figure 1: Schematic overview of the data and our modeling approach. **Left**: Data are observed across multiple time points and varying conditions, originating from a common source distribution (white), with "?" marking an unmeasured point of interest. **Middle**: Learning process. We train a neural network to regress the derivative (arrow) at points $x$ (red) sampled around the interpolation path (equipped with time-dependent variance, dashed gray lines). **Right**: Estimated trajectories after learning the underlying vector field – allowing the generation of paths for novel samples, imputing missing time points, and generalizing to unseen domains.

### 3.1 MULTI-MARGINAL FLOW MATCHING

We consider a collection of $K + 1$ discrete time stamps $\{t_0, \ldots, t_K\}$. Without loss of generality, we consider that the time stamps are distinct, ordered, and scaled so that $0 = t_0 < t_1 < \cdots < t_K = 1$. We don't impose any constraints on the spacing between consecutive time points. This flexibility accommodates scenarios where data collection occurs at irregular intervals, which is often the case in real-world applications. At each discrete time point $t_k$, we observe samples from a data density $p_{t_k}$ valued in $\mathcal{X}$.

**Flow Matching Objective**  We define $z := (x_0, \ldots, x_K) \in \mathcal{Z} = \mathcal{X}^{K+1}$ and for now, define $q$ as the data distribution with independent samples from each marginal. $q$ admits as density $q(z) = \Pi_{k=0}^{K} p_{t_k}(x_k)$. Applying Equation 3 with the augmented vector $z$ leads to the Multi-Marginal Flow Matching (MMFM) objective:

$$\mathcal{L}_{\text{MMFM}}(\theta) = \mathbb{E}_{t, z \sim q(z), x \sim p_t(x|z)} \left[ \|v_t(x; \theta) - u_t(x \mid z)\|_2^2 \right], \quad (7)$$

for any probability density path $p_t(x \mid z)$ and conditional vector field $u_t(x \mid z)$. This objective has the same form as Equation 3, but the density $q(z)$ and the probability density path are defined differently (over multiple time steps), resulting in a different target conditional vector field $u_t$. As demonstrated below, this remains a valid surrogate objective function for Flow Matching across multiple time points (the proof appears in Appendix A).

**Proposition 1.** *Assuming that $p_t(x) > 0$ for all $x \in \mathcal{X}$ and $t \in [0, 1]$, then, up to a constant independent of $\theta$, $\mathcal{L}_{FM}$ and $\mathcal{L}_{MMFM}$ are equal. Hence, for all values of the parameters $\theta$:*

$$\nabla_\theta \mathcal{L}_{FM}(\theta) = \nabla_\theta \mathcal{L}_{MMFM}(\theta). \quad (8)$$

**Conditional Probability Paths**  A key feature of MMFM is the specification of a Gaussian probability density path $p_t(x \mid z)$ that goes through all the time points $(x_0, \ldots, x_K)$. In contrast to classical FM, the MMFM framework can naturally incorporate prior knowledge about the system's dynamics over multiple time steps. For physical systems, a meaningful prior for $\mu_t$ is defined as an interpolating path of minimum energy or minimal curvature:

$$\mu_t(z) = \arg \min_{\gamma \in \mathcal{H}^2([t_0, t_K])} \int_{t_0}^{t_K} \|\gamma''(t)\|_2^2 \, dt \quad \text{s.t. } x_k = \gamma(t_k) \quad \text{for all } k \in \{0, \ldots, K\}, \quad (9)$$

where $\mathcal{H}^2([t_0, t_K])$ denotes the set of functions whose first derivative is absolutely continuous on $[t_0, t_K]$ and admit a weak second derivative. The celebrated Holladay's theorem (Holladay, 1957) demonstrates that $\mu$ is the natural cubic spline interpolation between the points $\{(x_k, t_k)\}_{k=0}^{K}$, whose coefficients can be computed efficiently since they only require solving a tri-diagonal system of linear equations.

Unlike the typical constant variance $\sigma_t$ employed in CFM, we also use a time-dependent variance function defined piece-wise on each time interval as

$$\sigma_t(z) = \frac{4(t_{k+1} - t)(t - t_k)}{(t_{k+1} - t_k)^2}, \quad (10)$$

for $t \in [t_k, t_{k+1}]$. Intuitively, this variance function adds noise to the vector field estimation in-between observed time points and is a crucial component for sharing information across conditions. Derivations for the temporal derivatives of $\mu_t(z)$ and $\sigma_t(z)$ are provided in Appendix D.

**Optimal Transport** Following the enhancements proposed by Tong et al. (2024) in the case with two marginals, we propose to sample data points across marginals using a joint distribution, computed as the solution of a multi-marginal optimal transport (MMOT) problem (Pass, 2015): $q(z) = \pi^*(x_0, \ldots, x_K)$. Notably, when the cost structure is pairwise additive for each pair of marginals, the MMOT problem reduces to a set of $K$ independent OT problems, such that $\pi^*(x_0, \ldots, x_K) = \frac{\prod_{k=1}^{K-1} \pi_k^*(x_k, x_{k+1})}{\prod_{k=2}^{K-1} p_{t_k}(x_k)}$, where $\pi_k^*(x_k, x_{k+1})$ is the solution of the OT problem between $p_{t_k}$ and $p_{t_{k+1}}$ (the proof is provided in Appendix B). Throughout this work, we assume such a pairwise additive cost structure. We pre-compute the solutions of the OT problems on the training data using the squared Euclidean distance as a cost function and use the multi-marginal optimal transport coupling for $q$ in Equation 7. When the dataset is too large, OT can be approximated using mini-batches, (Tong et al., 2024), although this was not necessary in our experiments.

**Relationship with CFM** MMFM is a generalization of CFM in the following sense. When the conditional vector field is defined with a piece-wise linear interpolation function for the mean and a constant variance, and in addition, the neural network $v_t$ is defined piece-wise on $[t_0, t_K]$, the MMFM problem is equivalent to solving $K$ distinct CFM problems between consecutive time points.

**Proposition 2.** *Let us assume that for all $z = (x_0, \ldots, x_K) \in \mathcal{Z}$, $\mu_t(z)$ is defined as the piecewise linear function going through all the points of $z$, and $\sigma_t(z) = \sigma$ is constant for all $z$. Additionally, let us assume that the vector field is learned separately on each time interval: $v_t(z; \theta) = \sum_{k=1}^{K} v_t(z; \theta_k) \mathbf{1}_{[t_k, t_{k+1}]}(t)$. Then, the MMFM problem is equivalent to solving $K$ separate CFM problems between each pair of consecutive marginals.*

The proof appears in Appendix A. This result highlights a key advantage of MMFM over multiple pairwise CFMs when the system's behavior exhibits similarities across time periods (or conditions, below). In such scenarios, MMFM can utilize a single set of parameters to model the dynamics across all time points (or, different conditions, below). This approach allows MMFM to leverage common patterns in the data that would be treated independently in separate pairwise CFM problems, leading to more efficient and robust modeling, and a deeper understanding of the underlying phenomena (e.g., biological commonalities).

## 3.2 CONDITIONAL MMOT-BASED MULTI-MARGINAL FLOW MATCHING

We now extend our framework to consider cases where each observation $x_k^c$ from time point $t_k$ is associated with a condition $c \in \{1, \ldots, C\}$. We assume the condition information is provided for each observation. We use the notation $x_k^c \sim p_{t_k}(x_k^c \mid c)$ to indicate the data density for condition $c$ at time point $t_k$. Additionally, we define $z^c = (x_0^c, \ldots, x_K^c)$. Importantly, our model can handle incomplete data scenarios, where not all conditions are observed at all time points. This extension enables us to capture and analyze condition-specific dynamics, such as the response of cells to different drugs in a chemical screen.

We leverage the conditional information in our architecture by (1) modifying the coupling $q(z)$ to be condition specific, and (2) extending to architecture of the vector field neural network $v_t$ to use the condition as input.

For the coupling, we use a condition-wise multi-marginal optimal transport coupling:

$$q^c(z^c) = \pi_c^*(x_0^c, \ldots, x_K^c) = \frac{\prod_{k=1}^{K-1} \pi_{c,k}^*(x_k^c, x_{k+1}^c)}{\prod_{k=2}^{K-1} p_{t_k}(x_k^c \mid c)}, \tag{11}$$

where $\pi_{c,k}^*$ is the optimal transport coupling between $p_{t_k}(x_k^c \mid c)$ and $p_{t_{k+1}}(x_{k+1}^c \mid c)$, and the second equality follows from the assumption of pairwise additive structure of the cost function (Section 3.1 and proof in Appendix B).

For the vector field neural network, we integrate the condition as an additional input to the neural model, modifying it to $v_t(x, c; \theta)$. This neural network takes as input a condition index $c$ that is used to internally retrieve a learnable condition embedding (details about the architecture are provided in

Appendix D). Importantly, we learn a set of shared weights $\theta$ across all conditions, which enables the model to generalize across different conditions and potentially improve performance on conditions with limited data.

To train the conditional model, we employ classifier-free guidance as proposed by Zheng et al. (2023). The objective for the condition-informed version of MMFM, termed *C-MMFM*, is defined as:

$$\mathcal{L}_{\text{C-MMFM}}(\theta) = \mathbb{E}_{t,b\sim\text{Ber}(p_{\text{u}}),c\sim\text{Cat}(C),z^c\sim\pi_c^*,x\sim p_t} \left\| v_t\left(x,(1-b)c+bc_\varnothing;\theta\right) - u_t(x \mid z^c) \right\|_2^2, \tag{12}$$

where $p_{\text{u}}$ represents the probability of switching to the unconditional model, and $c_\varnothing$ represents the null conditioning. The null conditioning $c_\varnothing$ is an additional condition index that retrieves a learnable embedding corresponding to a condition-independent model. When the Bernoulli variable $b$ is 1, the model uses a conditional-independent model. Otherwise, the conditional model is used. This approach allows information sharing across conditions via joint training of conditional $v_t(x,c;\theta)$ and unconditional $v_t(x,c_\varnothing;\theta)$ models.

At inference time, we use a weighted combination of the two vector fields (details about the sampling appear in Appendix C):

$$\tilde{v}_t(x,c;w,\theta) = (1-w)v_t(x,c_\varnothing;\theta) + wv_t(x,c;\theta) \tag{13}$$

and therefore can model the trajectory of a sample $x_0$ in condition $c$ as:

$$\hat{x}_T = x_0 + \int_0^T \tilde{v}_t(x,c;w,\theta)dt. \tag{14}$$

The parameters $p_{\text{u}}$ and $w$ are selected based on the value of the objective function on held-out validation data.

## 4 RELATED WORK

**Flow Matching**   Flow-based generative models have gained significant attention in recent years due to their ability to efficiently model generative processes for complex data distributions. Flow Matching, introduced by Lipman et al. (2023), proposed a simulation-free approach to train neural ODEs for generative modeling – a computationally attractive alternative to maximum likelihood training of continuous normalizing flows (Chen et al., 2018). Building upon this, Tong et al. (2024) extended FM by (1) allowing flows between arbitrary distributions (I-CFM), (2) proposing OT-CFM coupling samples using optimal transport theory (linking FM to dynamic optimal transport), and (3) introducing Schrödinger Bridge Conditional FM (SB-CFM), linking FM to Schrödinger Bridges. Kapusniak et al. (2024) improved FM by estimating the data manifold to adjust the linear interpolation path and ensuring it remains within high-density regions of the data. However, none of these methods takes into account multiple (more than two) time points.

**Label-Guided Flow Matching**   Zheng et al. (2023); Dao et al. (2023); Isobe et al. (2024) first integrated conditional information into FM. Guided Flows (Zheng et al., 2023) extended the FM framework to incorporate conditional information, allowing for more precise control over the generation process. At the same time, Dao et al. (2023) modeled the flows in a jointly learned latent space, offering improved computational efficiency. Meta Flow Matching (Atanackovic et al., 2024) extends traditional FM by modeling the flow over populations, *e.g.*, a set of cells from a patient, by embedding it using Graph Neural Networks. As with other Flow Matching studies, none of these methods consider multiple time points.

**Fast and Smooth Interpolation**   The approach in Chewi et al. (2021) shares similarities with MMFM in its use of OT principles to guide the interpolation process but differs in its specific implementation by directly using splines rather than neural network-based Flow Matching and cannot handle multiple conditions. Therefore, it also has the disadvantage of not being able to make inferences at test time using learned dynamics from other conditions.

**Multi Marginal Schrödinger Bridges** In contrast to Flow Matching, Schrödinger Bridge (SB) models, which are based on Stochastic Differential Equations (SDEs) and therefore predominantly estimated using Diffusion Models, have already been extended to make use of multiple marginals. The Deep Multi-Marginal Momentum Schrödinger Bridge (DMSB), introduced by Chen et al. (2023), extends the classical SB problem to scenarios with multiple marginal distributions by adjusting the Bregman iteration approach.

**Modeling of biological single-cell time series data** Understanding cellular mechanisms from biological time series data or *pseudo* time series data is a popular research area, as it helps understand fundamental dynamic processes in cells, from differentiation during development to pathogenesis (Schiebinger et al., 2019; Weinreb et al., 2020; Setty et al., 2019; Yeo et al., 2021; Qiu et al., 2022; Farrell et al., 2023; Bergen et al., 2020; Tong et al., 2020; Zhang et al., 2024). Advancements in single-cell profiling have enabled researchers to capture temporal snapshots of individual cell states across cell populations, from which computational methods can infer temporal trajectories and the underlying gene regulatory networks that control them. Despite the fact that temporal processes both span across multiple time points and are often shared across conditions, models to date have focused on either a single condition monitored along multiple time points (through modeling consecutive pairs) or on a series of conditions measured in a single time point.

## 5 EXPERIMENTS

We assessed the performance of MMFM using synthetic data as well as a single-cell RNA-seq dataset where each of multiple conditions (perturbations) was measured along multiple time points. We compared the performance of MMFM to that of several other methods, first considering methods explicitly designed to incorporate multiple time points. We considered FSI (Chewi et al., 2021), a method interpolating data densities using OT and natural cubic splines, and as an ablation study, we assessed the performance of a variant of MMFM with a piece-wise linear interpolation function (L-MMFM). We also sought to report the performance of DMSB (Chen et al., 2023) but did not see meaningful outputs after a long training time and, therefore, were not able to draw conclusions. We therefore considered more direct applications of the FM framework, applying OT-CFM (Tong et al., 2024) across consecutive pairs of marginals, and applied separately per condition (PCFM). Finally, we also compared to an OT-CFM model that would ignore all intermediate time points (CFM). We used three metrics for evaluation: Mean-Squared-Error (MSE), Maximum-Mean Discrepancy (MMD), and Wasserstein-2 ($W_2$) distance. The MSE provides a straightforward comparison of cluster centers, while MMD and $W_2$ offer more sophisticated assessments of distributional differences by considering higher-order moments and geometric properties. To ensure a fair comparison, we provided the condition-specific OT couplings as input to all models. Additional details about these experiments, including hyperparameter search grids and hold-out-based validation strategies, are provided in Appendix D. Considerations about computational complexity are given in Appendix J.

### 5.1 SYNTHETIC EXPERIMENTS

To evaluate MMFM's ability to estimate condition-specific vector fields from time course data, we created two-dimensional synthetic datasets with a known ground truth vector field. Data were generated for 12 conditions, defined as $c_m = m$ for $m \in \{1, \ldots, 12\}$ using the vector field

$$\begin{cases} u_t(x_t, c_m) = \begin{bmatrix} 3 \\ (\frac{c_m}{2} + 1)^{7/4} \cos(5\pi t) \end{bmatrix} \\[2mm] x_0(m) = \begin{bmatrix} 0, \dfrac{c_m - 2}{4} \end{bmatrix}. \end{cases} \tag{15}$$

We sampled from the time points and conditions described in Table 1. An overview of the data and the actual underlying trajectories appear in Figure 2A. We generated 50 data points per condition and time point according to the initial value problem above and included an additive Gaussian noise with variance $\sigma_c^2$. This scenario is similar to that in single-cell genomics, where the experimental assay is destructive, so each cell (sample) can be measured at most once. We held-out time point $t = 0.15$ during training for hyperparameter tuning of all FM-based models (but left it as input for the FSI model).

Table 1: Sampling times and conditions for the synthetic experiments.

| $t$ $c$ | 0 | 0.1 | 0.15 | 0.3 | 0.5 | 0.7 | 0.9 | 1.0 |
|---|---|---|---|---|---|---|---|---|
| $c_3$ | ✓ | ✓ | ✓ | – | ✓ | – | ✓ | ✓ |
| $c_5$ | ✓ | – | ✓ | ✓ | ✓ | ✓ | – | ✓ |
| Rest | ✓ | ✓ | ✓ | ✓ | ✓ | ✓ | ✓ | ✓ |

Note: ✓ indicates that samples from that combination are available in the training set.

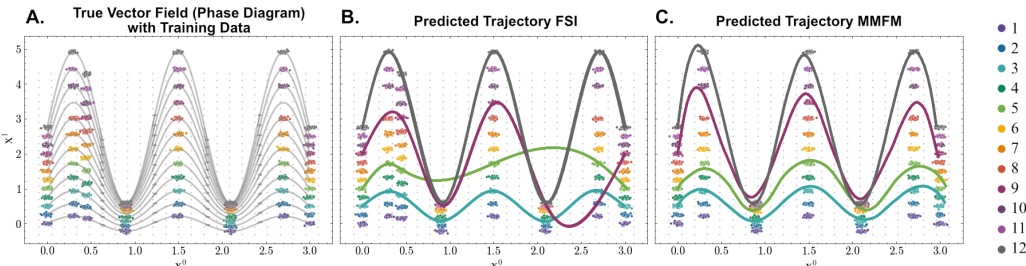

Figure 2: Assessment on synthetic data. (A) Ground truth vector field, which incorporates training samples for ten distinct conditions, each represented by a different color. (B,C) Interpolated paths generated by FSI (B) and MMFM (C) – illustrated for conditions $c \in [3, 5, 9, 12]$.

For evaluation, we compared deviations from the ground truth at the mean level, estimated with the Mean-Square Error (MSE), and at the distribution level, estimated by the Maximum Mean Discrepancy (MMD) between predicted samples and samples from the ground-truth vector field at 21 equidistant-spaced time points (from $t = 0$ to $t = 1$, by increments of $\Delta t = 0.05$).

The experimental results (Table 2) demonstrate the superior performance of MMFM on this time course data, as it most effectively predicts trajectories for conditions $c_3$ and $c_5$, where measurements are sparse and irregularly spaced. While FSI struggles to capture the right dynamics for the inferred trajectories (Figure 2) and systematically underestimated the number of inflection points, MMFM accurately adjusted the trajectories and aligned them with the curvature profile of other conditions (visualization of interpolation quality for other conditions and methods appear in Appendix E).

FSI and PCFM both cannot leverage information from other conditions, leading to incorrect trajectories and much higher errors. By contrast, CFM and MMFM have access to multiple conditions. However, CFM only considers the first and last time points, resulting in a higher loss than MMFM. Finally, L-MMFM performs substantially worse than MMFM, suggesting that the cubic splines are effective models for the dynamics of this dataset. Further results are provided in Appendix F, where we also discuss the extrapolation properties of MMFM.

Together, these results validate that MMFM is beneficial in scenarios where one has access to measurements taken at different time points and across different conditions.

Table 2: Results on synthetic data. We report the mean square error (MSE) and maximum mean discrepancy (MMD), with lower values indicating better performance. Means and standard deviations are computed over 21 time points. Best-performing models are highlighted in bold.

| | $c_m = 3$ | | $c_m = 5$ | |
|---|---|---|---|---|
| | MSE ↓ | MMD ↓ | MSE ↓ | MMD ↓ |
| FSI | 0.72 ± 0.60 | 0.42 ± 0.37 | 0.72 ± 0.93 | 0.33 ± 0.45 |
| CFM | 0.46 ± 0.25 | 0.25 ± 0.15 | 0.93 ± 0.52 | 0.60 ± 0.33 |
| PCFM | 0.75 ± 0.49 | 0.45 ± 0.35 | 0.78 ± 0.73 | 0.43 ± 0.44 |
| L-MMFM (ours) | 0.39 ± 0.25 | 0.20 ± 0.17 | 0.36 ± 0.27 | 0.17 ± 0.19 |
| MMFM (ours) | **0.13 ± 0.08** | **0.04 ± 0.02** | **0.22 ± 0.15** | **0.12 ± 0.08** |

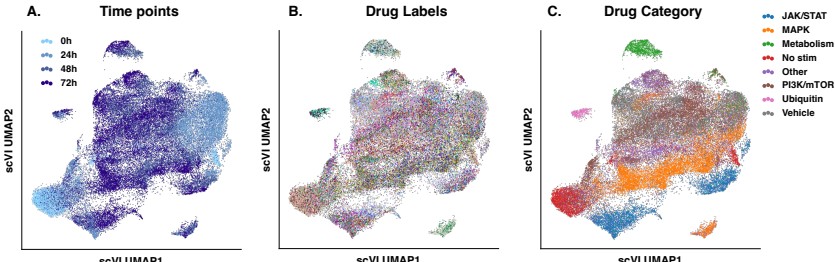

Figure 3: UMAP embeddings of single-cell RNA sequencing data visualizing drug responses over time. (A) Cells colored by time point (0h, 24h, 48h, 72h), highlighting the temporal progression of cellular states. (B) The same embedding colored by individual drugs (93 distinct labels), showcasing response heterogeneity. (C) Cells colored by drug categories, highlighting shared response patterns within drug classes. UMAP coordinates were derived using scVI (single-cell Variational Inference) (Lopez et al., 2018).

## 5.2 APPLICATION TO SINGLE-CELL PERTURBATION SCREENING DATA

We assessed the performance of MMFM on a real-world single-cell perturbation screen dataset. The dataset records single-cell gene expression ($18,250$ genes) profiles of T cells treated with kinase inhibitors and undergoing activation, measured at four time points (0h, 24h, 48h, and 72h). The dataset includes 93 distinct inhibitors, each used at varying concentrations (100 nM, 1 µM, and 10 µM), as well as negative control conditions (vehicle and non-activation). In each condition (combination of inhibitor, concentration and time point), hundreds of cells have been profiled. We treated the non-activation data as the $t = 0h$ time point for all conditions. To focus on the most significant effects, we filtered the data and retained $M = 123$ distinct treatments, each representing a unique combination of compound and concentration (Figure 3). We then randomly selected 60 of those treatments for analysis. For evaluation purposes, we withheld ten non-overlapping random treatments for each of the three time points. Standard single-cell data processing pipelines were used to normalize and scale the features, with the first 25 principal components used as input features (details regarding data preparation appear in Appendix H).

Because the ground-truth vector field is unknown in this setting, we evaluated the capacity of the methods to generalize to held-out time points. We thus assessed model performance at the mean level, estimated with the Mean-Square Error (MSE), and at the distribution level, estimated by the Wasserstein distance $W_2$ between predicted and hold-out samples. The training was conducted with five random seeds, using one held-out treatment to select the best model for testing. For this benchmark, the PCFM baseline was applied once using data from all conditions, as it otherwise required training 150 individual models, which was prohibitively time-consuming.

Table 3: Results for the drug response imputation task. We report the mean-square error (MSE) between the predicted and actual distributions' means and Wasserstein distance ($W_2$), where in both cases lower is better. Means and standard deviations are computed over five folds. Best-performing models are highlighted in bold.

| | $t = 24h$ | | $t = 48h$ | | $t = 72h$ | |
|---|---|---|---|---|---|---|
| | MSE ↓ | $W_2$ ↓ | MSE ↓ | $W_2$ ↓ | MSE ↓ | $W_2$ ↓ |
| FSI | 07.77 ± 01.49 | 56.13 ± 13.44 | 04.52 ± 01.14 | 38.93 ± 08.56 | 06.62 ± 01.65 | 78.46 ± 23.48 |
| CFM | 09.10 ± 01.59 | 60.86 ± 14.67 | 04.92 ± 01.88 | 33.28 ± 13.53 | – | – |
| PCFM | 08.88 ± 01.58 | 56.68 ± 12.12 | 04.13 ± 01.21 | 34.85 ± 08.01 | 06.07 ± 01.32 | 61.83 ± 16.74 |
| L-MMFM (ours) | 08.05 ± 01.29 | **50.82** ± 10.79 | 03.56 ± 01.50 | 26.07 ± 08.62 | 06.30 ± 01.92 | 47.97 ± 19.69 |
| MMFM (ours) | **07.54** ± 00.99 | 51.86 ± 07.84 | **03.38** ± 01.34 | **26.01** ± 07.72 | **04.92** ± 01.61 | **37.71** ± 12.24 |

Our experimental results (Table 3) demonstrate the superior performance of MMFM at imputing missing data across multiple time points. Interestingly, MMFM and L-MMFM have similar performance for the $t = 24h$ and $t = 48h$, but MMFM significantly outperforms L-MMFM at the $t = 72h$ time point. This suggests that the cubic spline is a more desirable prior for this modeling problem. In contrast, the reliance of PCFM on linear paths may introduce a bias towards straight-line trajectories, limiting its flexibility in capturing complex dynamics. Overall, MMFM demonstrates robust performance in handling complex datasets with missing temporal information, emphasizing its potential for diverse applications in generative modeling.

## 5.3 Application to Beijing Air Quality Data

To further study how well MMFM generalizes, especially under irregular sampling over time, we applied it to the Beijing multi-site air quality data set (Chen, 2017). This dataset comprises hourly air pollutant data from 12 air-quality monitoring sites across Beijing. We focused on PM2.5 data, which measures the density of particulate matter smaller than 2.5 micrometers, an important air pollution indicator, between January 2015 and February 2017. The 12 monitoring sites were modeled as the condition ($c \in \{1, \ldots, 12\}$). For each site, we grouped together the measurements collected over the same month, resulting in 26 temporal snapshots ($t \in \{1, \ldots, 26\}$). We selected different months as training data to simulate irregular sampling, with the interval between successive snapshots varying between 1 and 7 months. For 9 out of 12 stations, we selected 50% of the measurements, i.e. 13 months. For the other three stations, we selected only 7, 6 and 7 months as training data to simulate missing sensor data. These three stations are represented by the conditions $c = 4$, $c = 7$ and $c = 10$. We evaluated our method on all months that were not part of the training data set. This experiment represents a common challenge in sensor networks, where some measurements are lost or corrupted and must be imputed using data from other sensors. The results (Table 4) indicate that both MMFM versions (cubic spline and linear paths) show strong performance compared to FSI, CFM, and PCFM when it comes to interpolating the measurements. L-MMFM models show the best performance across all evaluated methods, as measured by MSE, MMFM is in second place in three out of four scenarios. Further details and evaluation metrics can be found in Appendix G.

Table 4: Results on the Beijing data. We report the maximum mean discrepancy (MMD), with lower values indicating better performance. Best-performing models are highlighted in bold.

|  | $c = 4$ | $c = 7$ | $c = 10$ | Rest |
|---|---|---|---|---|
| FSI | $1.70 \pm 0.40$ | $1.89 \pm 0.16$ | $1.92 \pm 0.10$ | $1.47 \pm 0.54$ |
| CFM | $1.86 \pm 0.19$ | $1.78 \pm 0.35$ | $1.85 \pm 0.21$ | $1.83 \pm 0.30$ |
| PCFM | $1.61 \pm 0.45$ | $1.42 \pm 0.36$ | $1.86 \pm 0.15$ | $1.50 \pm 0.50$ |
| L-MMFM (ours) | $\mathbf{1.33} \pm 0.61$ | $\mathbf{1.34} \pm 0.55$ | $\mathbf{1.24} \pm 0.68$ | $\mathbf{1.33} \pm 0.59$ |
| MMFM (ours) | $1.52 \pm 0.45$ | $1.50 \pm 0.49$ | $1.32 \pm 0.50$ | $1.52 \pm 0.51$ |

## 6 Discussion

We introduced Multi-Marginal Flow Matching (MMFM), a novel method for modeling complex system dynamics from temporal snapshot data across multiple conditions. We demonstrated that MMFM effectively combines data from various time points and experimental settings, outperforming existing methods in imputing missing time points. Our approach showed particular strength in scenarios with sparse or irregularly spaced measurements, leveraging information across conditions.

An important parameter in the design of MMFM is the time-varying variance function $\sigma_t$, as well as the sharing of parameters across conditions. An interesting direction for future work would be to add a set of parameters to the probability density path used in Flow Matching and provide a principled way to learn those parameters while simultaneously solving the FM problem. For example, one could use latent attentive neural processes (Kim et al., 2019) to construct meta-models across conditions. This approach could alleviate the need to find the right set of hyperparameters, ensuring that the conditioning network has sufficient, but not excessive, influence on the model parameters. Another promising avenue for constructing flows on real-world data is to explore more complex cost functions for multi-marginal optimal transport, (*e.g.*, the energy landscape-based loss functions for biological applications, Appendix I).

MMFM opens up new modeling opportunities for single-cell RNA-seq datasets. For example, so-called cellular velocities can be estimated from data either using nascent (unspliced) RNA, or metabolic labeling (La Manno et al., 2018; Bergen et al., 2020). Despite being inherently noisy, this additional information could be a valuable addition to enhance and steer the synthetic gradient flows based on interpolation paths, as previously demonstrated by Tong et al. (2020). Furthermore, constraining the model to satisfy specific stochastic differential equations, such as Ornstein-Uhlenbeck processes (Wang et al., 2023), could help bring interpretability, as well as mechanistic insights. Indeed, such models have analytical steady-state solutions and also accommodate for interventions (Rohbeck et al., 2024) (*e.g.,* modeling gene knock-outs). Lastly, biomedical data often includes various modalities providing multiple measurements on the same target (*e.g.,* a cell's gene expression and chromatin accessibility). Access to these multi-view datasets can allow for modeling coupled trajectories across different spaces (Somnath et al., 2023).

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

ACKNOWLEDGMENTS AND DISCLOSURES OF FUNDING

The authors would like to thank Maciek Wiatrak for fruitful discussions.

Disclosures: This work was done while Martin Rohbeck was an intern at Genentech. Romain Lopez, Edward De Brouwer, Charlotte Bunne, Anne Biton, Jan-Christian Huetter and Aviv Regev are employees of Genentech and/or have equity in Roche. Aviv Regev is a co-founder and equity holder of Celsius Therapeutics and an equity holder in Immunitas. She was an SAB member of ThermoFisher Scientific, Syros Pharmaceuticals, Neogene Therapeutics, and Asimov until July 31, 2020, and has been an employee of Genentech since August 1, 2020.

CODE AVAILABILITY STATEMENT

The code to reproduce the figures and tables, as well as to run the model and generate the simulated data, can be found at github.com/Genentech/MMFM.

DATA AVAILABILITY STATEMENT

Input data used for the experiments in this manuscript are currently undergoing publication as a separate manuscript. They will become publicly available upon acceptance.

APPENDICES

The following list gives an overview of the structure of the Appendix:

- In Appendix A, we present the proofs for the theoretical results behind our Multi-Marginal Flow Matching (MMFM) framework.
- In Appendix B, we go into the details of Multi-Marginal Optimal Transport (MMOT), explaining how it reduces to a sequence of pairwise Optimal Transport problems in our specific case.
- In Appendix C, we present the algorithm for sampling from the COT-MMFM model.
- In Appendix D, we outline our model training and evaluation approach, including hyper-parameter tuning strategies.
- In Appendix E, we provide additional experimental details and visualizations for our synthetic data experiments.
- In Appendix F, we showcase an additional synthetic experiment focused on the extrapolation capabilities of our MMFM model.
- In Appendix G, we provide further details on preprocessing and training-validation-test splits on the Beijing weather data set.
- Appendix H describes the preprocessing steps and treatment selection criteria for the perturbation screening data used in our real-world experiments.
- In Appendix I, we offer a deeper motivation for using the minimum curvature prior as a proxy for energy minimization along a trajectory.
- Finally, in Appendix J, we provide a detailed analysis on the complexity of the method including the optimal transport step.

## A  PROOF OF PROPOSITIONS

**Proposition 1.** *Assuming that $p_t(x) > 0$ for all $x \in \mathcal{X}$ and $t \in [0, 1]$, then, up to a constant independent of $\theta$, $\mathcal{L}_{FM}$ and $\mathcal{L}_{MMFM}$ are equal. Hence, for all values of the parameters $\theta$:*

$$\nabla_\theta \mathcal{L}_{FM}(\theta) = \nabla_\theta \mathcal{L}_{MMFM}(\theta). \tag{8}$$

*Proof.* Following Lipman et al. (2023), we begin by constructing the target probability path $p_t(x)$ from the conditional probability path $p_t(x \mid z)$. Given a particular draw from the sample distribution

$z = (x_0, \ldots, x_K)$, we design the conditional probability path $p_t(x \mid z)$ such that it satisfies the following conditions:

$$\forall k \in \{0, \ldots, K\}, \ p_{t_k}(x \mid z) = \text{Normal}\left(x_k, \sigma^2 I\right), \tag{16}$$

where $\sigma > 0$ is small so that $p_{t_k}(x \mid z)$ is well concentrated around $x_k$. Marginalizing against the distribution $q(z)$, we obtain the marginal probability path:

$$p_t(x) = \int p_t(x \mid z) q(z) dz. \tag{17}$$

For each time point $t_k$, the marginal probability $p_{t_k}$ closely approximates the $k$-th marginal:

$$p_{t_k}(x) = \int f\left(\frac{x'_k - x_k}{\sigma}\right) q_{t_k}(x'_k) dx'_k \xrightarrow[\sigma \to 0]{} q_{t_k}(x_k), \tag{18}$$

where $f$ is the density of the $d$-dimensional isotropic Gaussian distribution.

Similarly to the concept of marginal probability paths, we can also introduce the marginal vector field (assuming $p_t(x) > 0$ for all $x \in \mathcal{X}$):

$$u_t(x) = \int u_t(x \mid z) \frac{p_t(x \mid z)}{p_t(x)} q(z) dz, \tag{19}$$

where $u_t(x \mid z)$ is the conditional vector field that generates $p_t(x \mid z)$. This is guaranteed by applying the proof of Theorem 1 from Lipman et al. (2023).

With this proper setup, the remainder of the proof follows closely the one from Theorem 2 in Lipman et al. (2023). We reformulate it below for completeness.

We assume that $q, p_t(x \mid z)$ are decreasing to zero at sufficient speed as $\|x\| \to \infty$ and that $u_t, v_t, \nabla_\theta v_t$ are bounded. We start to derive from the left-hand side:

$$\nabla_\theta \mathcal{L}_{\text{FM}}(\theta) = \nabla_\theta \mathbb{E}_{p_t(x)} \|v_t(x; \theta) - u_t(x)\|^2$$

$$\overset{\dagger}{=} \nabla_\theta \mathbb{E}_{p_t(x)} \left( \|v_t(x; \theta)\|^2 - 2\langle v_t(x; \theta), u_t(x) \rangle + \|u_t(x)\|^2 \right)$$

$$\overset{*}{=} \nabla_\theta \mathbb{E}_{p_t(x)} \left( \|v_t(x; \theta)\|^2 - 2\langle v_t(x; \theta), u_t(x) \rangle \right)$$

and rewrite the right-hand side:

$$\nabla_\theta \mathcal{L}_{\text{MMFM}}(\theta) = \nabla_\theta \mathbb{E}_{q(z), p_t(x|z)} \|v_t(x; \theta) - u_t(x \mid z)\|^2$$

$$\overset{\dagger}{=} \nabla_\theta \mathbb{E}_{q(z), p_t(x|z)} \left( \|v_t(x; \theta)\|^2 - 2\langle v_t(x; \theta), u_t(x \mid z) \rangle + \|u_t(x \mid z)\|^2 \right)$$

$$\overset{*}{=} \mathbb{E}_{q(z), p_t(x|z)} \nabla_\theta \left( \|v_t(x; \theta)\|^2 - 2\langle v_t(x; \theta), u_t(x \mid z) \rangle \right),$$

where we drop all terms independent of $\theta$ (*) and use the bilinearity of the 2-norm (†).

First, we show that the first terms of both equations are equal.

$$\mathbb{E}_{p_t(x)} \|v_t(x; \theta)\|^2 = \int \|v_t(x; \theta)\|^2 p_t(x) dx$$

$$= \int \|v_t(x; \theta)\|^2 \left( \int p_t(x \mid z) q(z) dz \right) dx$$

$$\overset{*}{=} \int \int \|v_t(x; \theta)\|^2 p_t(x \mid z) q(z) dz dx$$

$$\overset{\dagger}{=} \mathbb{E}_{q(z), p_t(x|z)} \|v_t(x; \theta)\|^2,$$

where we use Fubini's theorem to change the order of the integrals (*), and the law of total expectation (†).

Second, we show the equality of the latter parts. To get started, we plug the definition from Equation 19 into the term of interest:

$$\mathbb{E}_{p_t(x)} \langle v_t(x;\theta), u_t(x) \rangle = \int \left\langle v_t(x;\theta), \int \frac{u_t(x \mid z)p_t(x \mid z)q(z)dz}{p_t(x)} \right\rangle p_t(x)dx. \quad (20)$$

Thanks to the linearity of the integral, we may take the inner integral outside of the inner product:

$$\mathbb{E}_{p_t(x)} \langle v_t(x;\theta), u_t(x) \rangle = \int \int \left\langle v_t(x;\theta), \frac{u_t(x \mid z)p_t(x \mid z)q(z)}{p_t(x)} \right\rangle p_t(x)dzdx. \quad (21)$$

Employing the linearity of the inner product, we may rearrange the terms as:

$$\mathbb{E}_{p_t(x)} \langle v_t(x;\theta), u_t(x) \rangle = \int \int \langle v_t(x;\theta), u_t(x \mid z) \rangle q(z)p_t(x \mid z)dzdx. \quad (22)$$

Finally, using Fubini's theorem to change the order of the integrals and the law of total expectation, we recognize the following term:

$$\mathbb{E}_{p_t(x)} \langle v_t(x;\theta), u_t(x) \rangle = \mathbb{E}_{q(z),p_t(x|z)} \langle v_t(x;\theta), u_t(x \mid z) \rangle, \quad (23)$$

which concludes our proof. $\qquad \square$

**Proposition 2.** *Let us assume that for all* $z = (x_0, \ldots, x_K) \in \mathcal{Z}$, $\mu_t(z)$ *is defined as the piecewise linear function going through all the points of* $z$, *and* $\sigma_t(z) = \sigma$ *is constant for all* $z$. *Additionally, let us assume that the vector field is learned separately on each time interval:* $v_t(z;\theta) = \sum_{k=1}^{K} v_t(z;\theta_k)\mathbf{1}_{[t_k,t_{k+1})}(t)$. *Then, the MMFM problem is equivalent to solving* $K$ *separate CFM problems between each pair of consecutive marginals.*

*Proof.* We remind the reader of the expression of the MMFM optimization problem:

$$\mathcal{L}_{\text{MMFM}}(\theta) = \mathbb{E}_{t,z\sim q(z),x\sim p_t(x|z)} \left[ \|v_t(x;\theta) - u_t(x \mid z)\|_2^2 \right]. \quad (24)$$

We first notice that this time integral from 0 to 1 may be broken up into each of the segments in between observed marginal distributions:

$$\mathcal{L}_{\text{MMFM}}(\theta) = \sum_{k=0}^{K-1} I_k, \quad (25)$$

where we define the integral $I_k$ as:

$$I_k = \int_{t_k}^{t_{k+1}} \mathbb{E}_{z\sim q(z),x\sim p_t(x|z)} \left[ \|v_t(x;\theta) - u_t(x \mid z)\|_2^2 \right] dt. \quad (26)$$

The main goal of this proof is to see whether the integrand of each $I_k$ depends only on information from the marginals $q_{t_k}$ and $q_{t_{k+1}}$ and can be related to a Flow Matching problem.

Because $\mu_t(z)$ is the piecewise interpolation between all the points $z = (x_0, \ldots, x_K)$ at time $(t_0, \ldots, t_K)$, it has the closed-form expression:

$$\mu_t(z) = \sum_{k=0}^{K-1} \left( x_k + \frac{t - t_k}{t_{k+1} - t_k}(x_{k+1} - x_k) \right) \cdot \mathbf{1}_{[t_k,t_{k+1})}(t), \quad (27)$$

where $\mathbf{1}_{[t_k,t_{k+1})}(t)$ is the indicator function, which equals 1 when $t \in [t_k, t_{k+1})$ and 0 otherwise.

Using these, we may plug in the expressions for $\mu_t(z)$ and $\sigma_t(z)$ into the formula for the conditional vector field:

$$u_t(x \mid z) = \sum_{k=0}^{K-1} \frac{x_{k+1} - x_k}{t_{k+1} - t_k} \mathbf{1}_{[t_k,t_{k+1})}(t). \quad (28)$$

Finally, plugging in the piece-wise formula for $v_t$, this is enough to identify that:

$$I_k = \int_{t_k}^{t_{k+1}} \mathbb{E}_{z\sim\pi(x_k,x_{k+1}),x\sim\tilde{p}_t(x|x_k,x_{k+1})} \left\| v_t(x;\theta_k) - \frac{x_{k+1} - x_k}{t_{k+1} - t_k} \right\|_2^2 dt, \quad (29)$$

where $\pi$ is the result of marginalizing $q$ with respect to all variables, except $x_k$ and $x_{k+1}$, and $\tilde{p}_t(x \mid x_k, x_{k+1})$ is a probability path linearly interpolating the means between $x_k$ and $x_{k+1}$, and with constant variance. Therefore, it is a CFM problem between two data distributions, as in Tong et al. (2024).

Because each $I_k$ does not depend on other marginals than at time $k$ and $k+1$, and the neural network $v_t(x; \theta_k)$ is only learned on the interval $[t_k, t_{k+1}]$, then optimizing the sum of each $I_k$ becomes equivalent to optimizing each $I_k$ independently. Therefore, the MMFM problem is equivalent to $K$ separate CFM problems. $\qquad\square$

## B  MULTI-MARGINAL OPTIMAL TRANSPORT

Multi-marginal optimal transport (MMOT) (Pass, 2015) extends the classical Optimal Transport (OT) problem to multiple probability measures, aiming to find a coupling that minimizes the total cost associated with transporting mass between multiple marginals.

The optimization problem for $K$ marginals can be expressed considering all potential couplings between measures $\mu_1, \ldots, \mu_K$ supported on the same metric space $(\mathcal{X}, \|\cdot\|_2)$. For the cost function $c : \mathcal{X}^K \to \mathbb{R}$, the MMOT objective is:

$$\inf_{\pi \in \Pi(\mu_1, \ldots, \mu_K)} \int_{\mathcal{X}^K} c(x_1, \ldots, x_K) \, d\pi(x_1, \ldots, x_K), \tag{30}$$

where $\Pi(\mu_1, \ldots, \mu_K)$ is the set of probability measures over $\mathcal{X}^K$ with marginals equal to $\mu_1, \ldots, \mu_K$. For $K = 2$, this reduces to the classical two-marginal optimal transport problem.

Potentially, this could be a great opportunity to constrain the matching across multiple time points to be physically realistic, for example, by minimizing the curvature of the transport across time. In the case of cellular biology, one might expect that cells strive to maintain homeostasis and minimize changes across time.

For the sake of simplicity, we focus on pairings between consecutive marginals (with respect to time), and define a cost function that is linear across pairs of consecutive spaces. We demonstrate that this approach reduces the MMOT problem to a sequence of pairwise OT problems. We refer the reader to (Haasler et al., 2021) for the general case.

In our case, the MMOT problem becomes:

$$\inf_{\pi \in \Pi(\mu_1, \ldots, \mu_K)} \int_{\mathcal{X}^K} \left[ \sum_{k=1}^{K-1} c^k(x_k, x_{k+1}) \right] d\pi(x_1, \ldots, x_K), \tag{31}$$

which, after permutation of the integral and the discrete sum, yields:

$$\inf_{\pi \in \Pi(\mu_1, \ldots, \mu_K)} \sum_{k=1}^{K-1} \iint_{\mathcal{X} \times \mathcal{X}} c^k(x_k, x_{k+1}) d\pi(x_k, x_{k+1}). \tag{32}$$

Let us now denote by $\pi_k^*(x_k, x_{k+1})$ the solution of the OT problem with cost $c^k$ for each $k \in \{1, \ldots, K-1\}$. We consider the composite density:

$$\pi^*(x_1, \ldots, x_K) = \frac{\prod_{k=1}^{K-1} \pi_k^*(x_k, x_{k+1})}{\prod_{k=2}^{K-1} \mu_k(x_k)}. \tag{33}$$

Because $\pi_k^*$ is an optimal transport plan for each of the $k$ elements in the sum above, it must be that for all $\pi \in \Pi(\mu_1, \ldots, \mu_K)$:

$$\sum_{k=1}^{K-1} \iint_{\mathcal{X} \times \mathcal{X}} c^k(x_k, x_{k+1}) d\pi_k^*(x_k, x_{k+1}) \leq \sum_{k=1}^{K-1} \iint_{\mathcal{X} \times \mathcal{X}} c^k(x_k, x_{k+1}) d\pi(x_k, x_{k+1}). \tag{34}$$

If we can show that (1) the marginals of $\pi^*$ correspond to the intended marginals, that is $\pi^* \in \Pi(\mu_1, \ldots, \mu_K)$ and (2) the pairwise marginals of $\pi^*$ correspond to the solution of the pairwise

OT, then the lower bound in Equation 34 is tight. Consequently, $\pi^*$ solves the MMOT problem; therefore, the solution of MMOT can be achieved by solving a sequence of OT problems. These steps of the proof rely on the specific assumption about the form of the marginals.

We present the example for $K = 3$ without loss of generality (the main proof can be written by induction on $K$). We have that:

$$\pi^*(x_1, x_2, x_3) = \frac{\pi_1^*(x_1, x_2)\pi_2^*(x_2, x_3)}{\mu_2(x_2)} \tag{35}$$

In that case, the pairwise marginal of $\pi^*(x_2, x_3)$ is evaluated as:

$$\pi^*(x_2, x_3) = \frac{\pi_2^*(x_2, x_3)}{\mu_2(x_2)} \left[ \int_{\mathcal{X}} \pi_1^*(x_1, x_2)dx_1 \right] = \pi_2^*(x_2, x_3), \tag{36}$$

where the second equality holds because $\pi_1^*$ is an optimal transport map, we have that it satisfies the marginal condition. Together, this verifies the equality between the pairwise marginals of $\pi^*$ and the solutions of pairwise OT by the symmetry of the roles played by $x_1$, $x_2$, and $x_3$. Because the pairwise marginals are the OT transport plan, we must have that $\pi^* \in \Pi(\mu_1, \ldots, \mu_K)$, which completes the proof.

## C  METHOD DETAILS

We present the detail for the sampling of trajectories from the MMFM model in Algorithm 1. To generate test trajectories from the vector fields, we employed the fourth order Runge-Kutta method (*rk4*). Note that the guidance strength $w$ can be greater than 1 in this formulation.

---

**Algorithm 1** Pseudocode: Sampling from COT-MMFM

---

**Require:** Trained $v_t(x, c; \theta)$, guidance parameter $w$, condition $c$, no. of steps $T$, sample $x_{t=0}$
1: Set: t = 1/T
2: Define: $\tilde{v}_t(x, c; \theta) = (1 - w) \cdot v_t(x, c = \varnothing; \theta) + w \cdot v_t(x, c; \theta)$
3: **for** k = 1 to $T$ **do**
4:    $x_{k+1} = \text{ODEIntegrator}(x_k, \tilde{v}_t(x, c; \theta), t)$
5: **end for**
6: **return** $x_T$

---

## D  MODEL TRAINING AND EVALUATION

To ensure a fair comparison across all models, we constructed the data sets for each model based on the need for validation data and conducted each experiment at least three times with varying seeds. For models necessitating hyperparameter tuning, we employed a hold-out train-validation split of the samples, selecting the model that achieved the best performance on the validation set. In contrast, for FSI, where hyperparameter tuning was not required, we used the entire data set for training, including the hold-out part. Tab. 5 provides an overview of the hyperparameters tuned for each MMFM model.

**Architecture of $v_t(x, c; \theta)$.**  Given an observation $x$, a condition $c$, and a time $t$, we first compute a temporal embedding for $t$ using a sinusoidal timestep embeddings (Vaswani et al., 2017). We then retrieve a condition embedding from a library of learnable condition embeddings, using the index $c$. The input $x$ is transformed into a representation via a three-layer feedforward network using LeakyReLU activations. Subsequently, the embeddings of $x$, $t$, and $c$ are concatenated and processed through an additional three-layer neural network. The model is optimized using the Adam optimizer (Kingma & Ba, 2015).

**Mean and Variance Function**  The definition of $\mu_t(z)$ can be found in Equation 9 and represents the natural cubic spline-based mean-function of the flow. Hence, the mean function $\mu_t(z)$ is a piecewise third-order polynomial function with $\mathcal{C}^2$ continuity, meaning its values and first and

second derivatives are continuous at interval boundaries. For each interval $[t_k, t_{k+1}]$, the spline can be expressed as:

$$\mu(t, z) = a_k(t - t_k)^3 + b_k(t - t_k)^2 + c_k(t - t_k) + d_k,$$

where $a_k$, $b_k$, $c_k$, and $d_k$ are coefficients specific to that interval. Therefore, the first derivative is:

$$\mu^{'}(t, z) = 3a_k(t - t_k)^2 + 2b_k(t - t_k) + c_k.$$

The spline and its derivative can be efficiently computed by solving a tridiagonal system of linear equations. For the variance function $\sigma_t(z)$, we considered two options: (1) Constant values and therefore $\sigma_t^{'}(z) = 0$ or (2) if $\sigma_t(z)$ is as defined in Equation 10, then we have $\sigma_t'(z) = 4 \cdot \frac{t_k + t_{k+1} - 2t}{(t_{k+1} - t_k)^2}$. Both of these formulas are also efficient to compute.

**Data Preprocessing** We precomputed optimal couplings between samples of consecutive time steps in all synthetic experiments. This approach eliminated the need for mini-batching in each iteration, thereby enhancing computational efficiency.

Table 5: Hyperparameters for model training. (*) Applicable to all variations of Flow Matching models discussed in the paper.

| Model | Hyperparameters | Values/Range |
|---|---|---|
| FSI | - | - |
| MMFM* | learning rate | [1e-2, 1e-3, 1e-4] |
| | $p_{\mathrm{u}}$ | [0.0, 0.1, 0.2, 0.3] |
| | latent dimensions $(x, t, c)$ | [16, 32, 64, 128, 256] |
| | flow variance | [0.01, 0.1, 1, adaptive] |
| | guidance $w$ | $[\frac{k}{10}]$ for $k \in \{1, \ldots, 10\} \cup \{20, 30\}$ |

We provide two distinct approaches for integrating conditional information into the neural network model. The first approach was used throughout this study. This second approach is particularly advantageous when dealing with ordinal condition types, such as drug dosage, allowing to incorporate relationships between embeddings into the model.

**Learnable Embeddings:** This approach learns separate embeddings for each condition and the null condition $\varnothing$. While yielding superior results compared to the second approach (see below), it does not allow for inter-/extrapolation to new conditions. However, when data for a previously unseen condition becomes available, the model can be fine-tuned by fixing the weights of $v_t(x, c; \theta)$ and re-training only the new embedding. This approach generally necessitates a separate hold-out distribution to select the most promising model for test scenarios.

**Fixed Embeddings:** This approach either (i) concatenates the value of the condition directly to the input vectors $x$ and $t$ or (ii) before concatenating uses another neural network to compute embeddings for each condition, which are then concatenated to the input, e.g., $e(c) : \mathbb{R}^m \to \mathbb{R}^n$. The encoder weights are then learned during training, allowing the model to generalize to unseen conditions without model retraining.

**Optimal Transport Implementation** As shown in Appendix B, the multi-marginal optimal transport map decomposes as a product on two-marginal transport maps $\pi_k^*(x_k, x_{k+1})$. Practically, we solve the above problem for each $(t_k, t_{k+1})$ and $c$ using the *ot.emd2* function provided in the Python package POT.

# E EXPERIMENTAL DETAILS ON SYNTHETIC DATA

The following figures illustrate the condition-specific trajectory prediction of MMFM, L-MMFM, and the FSI model. Fig. 5 indicates that linear interpolation generally leads to straighter paths. However, the model can correct the paths for $c_3$ and $c_5$. Fig. 6 shows that FSI is able to match all marginals perfectly, which leads to vastly incorrect trajectories for $c_3$ and $c_5$.

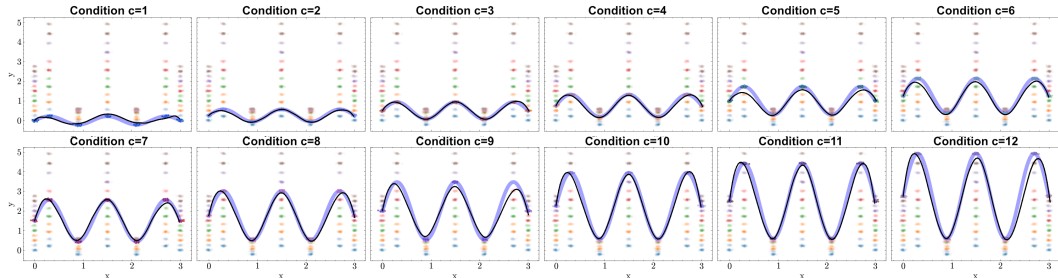

Figure 4: Visualization of MMFM interpolation results across all experimental conditions. The ground truth vector field is shown in blue, with predicted test sample trajectories in black. Training samples are displayed in the background using transparent, colored markers.

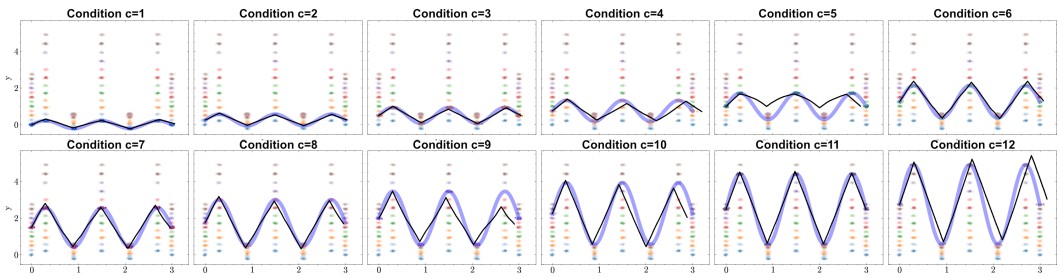

Figure 5: Visualization of L-MMFM interpolation results across all experimental conditions. The ground truth vector field is shown in blue, with predicted test sample trajectories in black. Training samples are displayed in the background using transparent, colored markers.

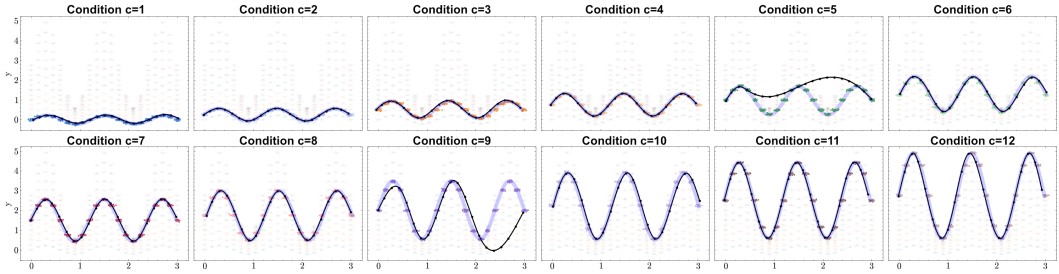

Figure 6: Visualization of FSI interpolation results across all experimental conditions. The ground truth vector field is shown in blue, with predicted test sample trajectories in black. Training samples are displayed in the background using transparent, colored markers.

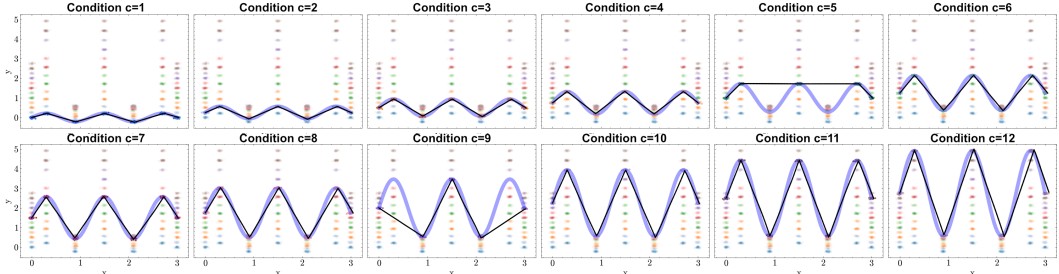

Figure 7: Visualization of PCFM interpolation results across all experimental conditions. The ground truth vector field is shown in blue, with predicted test sample trajectories in black. Training samples are displayed in the background using transparent, colored markers.

### E.1 Extrapolation To Unseen Starting Points

We evaluate the extrapolation capabilities of our model by extending the estimated vector fields to unseen regions for fixed conditions, as illustrated in Fig. 8. The results demonstrate that the underlying vector fields exhibit smooth behavior in the neighborhoods surrounding the interpolated paths (highlighted in blue). However, when initializing trajectories from unexplored regions, i.e., regions without training data being provided to the model beforehand, specifically where $y < 0$, we observe a tendency for the trajectories to overshoot beyond $x = 3$. Furthermore, for condition $c_3$, we note that the trajectories are influenced by the probability path derived from the spline, which was constrained to only five time points and followed a linear trajectory, and therefore divides the vector field into two regions (above and below the linear, horizontal line). This highlights the impact of the interpolation method on the model's extrapolation performance.

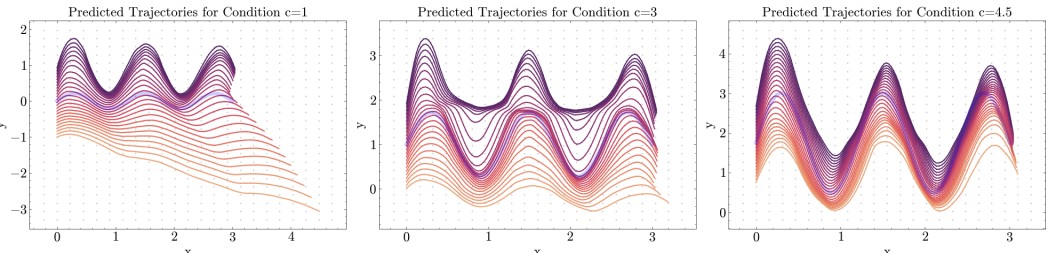

Figure 8: Visualization of MMFM inter-/extrapolation using unseen starting points across three conditions. The ground truth vector field is shown in blue, overlaid with the predicted test sample trajectories.

## F Additional Synthetic Experiment: Extrapolation to unseen Time Points

Next, we evaluated the extrapolation capabilities of our MMFM model to unseen time points for some conditions. Therefore, we generated data samples from a sinusoidal curve at time points $t = 0, 0.25, 0.5, 0.75, 1$ across conditions $c = 1, \ldots, 10$. For condition 5, the model was provided with measurements only at $t = 0, 0.25, 0.5$. Figure 9 shows the interpolated trajectories for test samples. The findings demonstrate that MMFM effectively extrapolates the time course data for $c_5$ by leveraging a smooth vector field constructed based on the other conditions. In contrast, as expected, FSI fails to predict future behavior accurately despite having access to an additional data point at $t = 0.55$ during training.

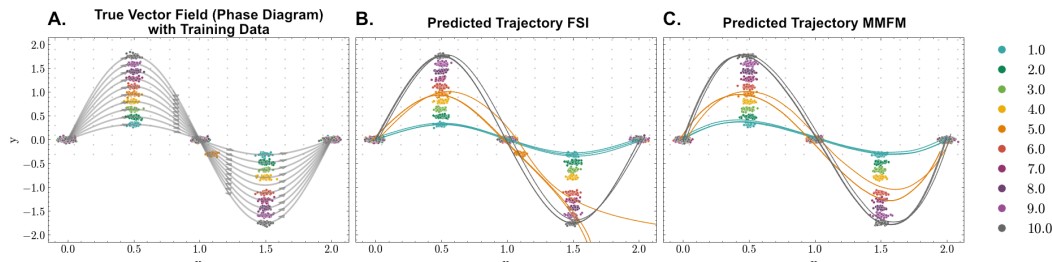

Figure 9: Overview of extrapolation to time points. (A) Visualization of training samples, (B) predicted trajectories for FSI, and (C) predicted trajectories for MMFM. Trajectories are shown for $c_1, c_5, c_{10}$. Here $c_5$ is only measured until $t = 0.5$.

## G Details about the Beijing Weather Data

**Data Preprocessing** The data was downloaded from archive.ics.uci.edu. We selected the PM2.5 concentration as the dependent variable for our analysis. Next, we grouped the data by month and computed the mean PM2.5 values.

For training, we used the subset of months as shown in Table 6 and Table 7 (months enumerated

from 0 (January 2015) to 25 (February 2017)). For testing, we used all other time points that were not part of the training. For the MMFM methods, we used month 11 for condition $c_4$ for validation and provided this data as additional training data to FSI.

Table 6: Sampling times (month 0-12) and conditions for the Beijing weather experiment.

| $c$ \ $t$ | 0 | 1 | 2 | 3 | 4 | 5 | 6 | 7 | 8 | 9 | 10 | 11 | 12 |
|---|---|---|---|---|---|---|---|---|---|---|---|---|---|
| $c_4$ | ✓ | - | - | - | - | ✓ | - | - | - | ✓ | ✓ | - | - |
| $c_7$ | ✓ | - | - | ✓ | - | - | - | - | - | - | - | - | - |
| $c_{10}$ | ✓ | - | - | - | ✓ | - | - | - | - | - | - | ✓ | - |
| Rest | ✓ | - | ✓ | - | ✓ | - | ✓ | ✓ | - | - | - | ✓ | ✓ |

Note: ✓ indicates that samples from that combination are available in the training set.

Table 7: Sampling times (month 13-25) and conditions for the Beijing weather experiment.

| $c$ \ $t$ | 13 | 14 | 15 | 16 | 17 | 18 | 19 | 20 | 21 | 22 | 23 | 24 | 25 |
|---|---|---|---|---|---|---|---|---|---|---|---|---|---|
| $c_4$ | - | - | - | - | ✓ | - | - | ✓ | - | - | - | - | ✓ |
| $c_7$ | ✓ | - | - | - | ✓ | - | - | - | - | - | ✓ | - | ✓ |
| $c_{10}$ | ✓ | - | - | - | ✓ | - | - | - | - | - | ✓ | - | ✓ |
| Rest | - | ✓ | - | ✓ | - | ✓ | ✓ | - | - | - | ✓ | - | ✓ |

Note: ✓ indicates that samples from that combination are available in the training set.

Table 8: Results on the Beijing data. We report the Mean-Squared Error (MSE), with lower values indicating better performance. Best-performing models are highlighted in bold.

| | $c = 4$ | $c = 7$ | $c = 10$ | Rest |
|---|---|---|---|---|
| FSI | 29.54 ± 30.99 | 45.31 ± 34.05 | 53.42 ± 35.53 | 17.20 ± 20.11 |
| CFM | 28.18 ± 18.60 | 20.22 ± 12.43 | 28.79 ± 18.46 | 26.86 ± 16.86 |
| PCFM | 18.47 ± 16.42 | 6.87 ± 5.37 | 35.18 ± 23.27 | 12.77 ± 11.10 |
| L-MMFM (ours) | **9.73** ± 9.33 | **6.33** ±7.68 | **9.03** ± 9.02 | **9.21** ± 8.90 |
| MMFM (ours) | 11.43 ± 9.05 | 13.74 ± 14.44 | 11.41 ± 17.07 | 17.02 ±18.72 |

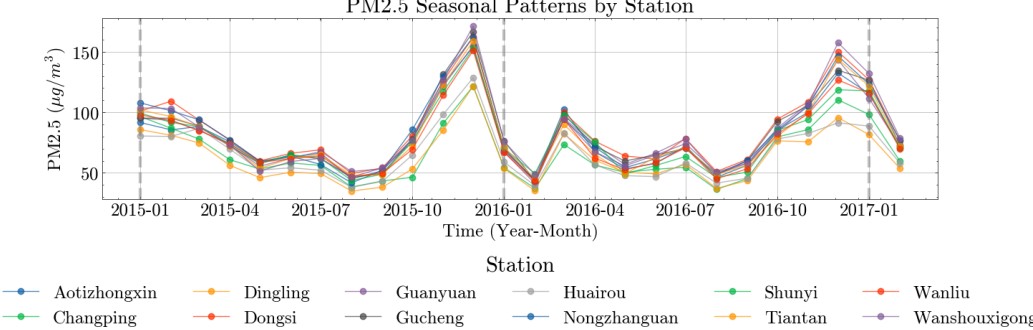

Figure 10: Overview of the preprocessed Beijing data set.

## H DETAILS ABOUT THE PERTURBATION SCREENING DATA

**Treatment selection** We implemented a two-stage approach to eliminate treatments with minimal effects. First, we trained a scVI model (Lopez et al., 2018) on the full count dataset using 50 latent features and a negative binomial likelihood. We then calculated the energy distance between the control group and each treatment in the latent space, keeping treatments with an energy distance above a certain threshold, here $\tau = 1$. As described in the main text, we randomly selected a subset of treatments and sampled the hold-out data for our experiments also at random. Non-stimulated control and vehicle data were also preserved for the experiments.

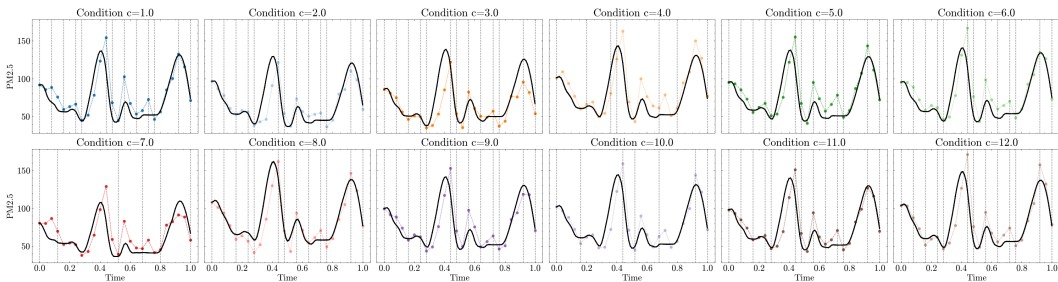

Figure 11: Visualization of MMFM interpolation results across all stations. The vertical bars indicate the training time points.

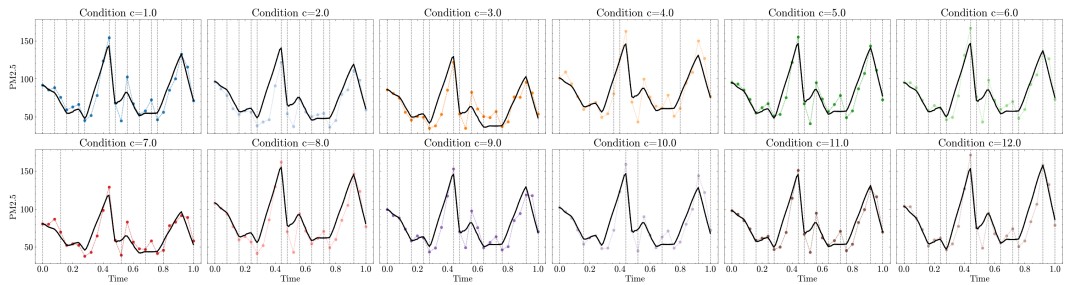

Figure 12: Visualization of L-MMFM interpolation results across all stations. The vertical bars indicate the training time points.

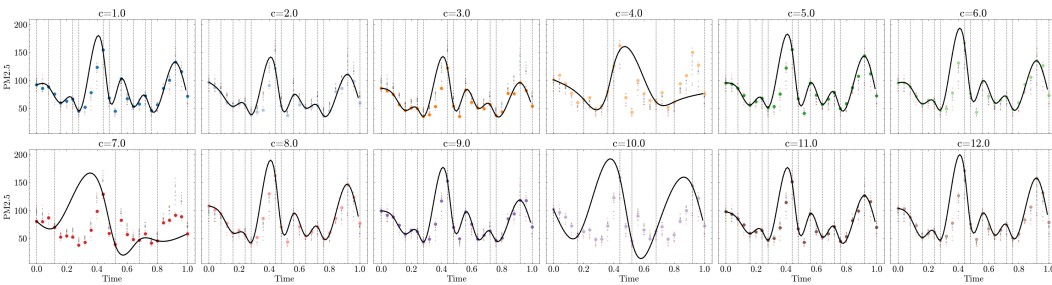

Figure 13: Visualization of FSI interpolation results across all stations. The vertical bars indicate the training time points.

**Data Preprocessing** To train the models, the mRNA count data was normalized for library size to remove technical noise and $\log(x+1)$-transformed to stabilize the variance and reduce the impact of extreme values. Following normalization and transformation, we applied Principal Component Analysis (PCA) to reduce the dimensionality of the 18,500-dimensional gene feature space. We retained the top 25 principal components, capturing most of the data's variance.

## I  PRIORS FOR TRAJECTORIES DYNAMICS

In this section, we further motivate the minimum curvature prior as a proxy for energy minimization along a trajectory. We consider the trajectories of a particle of initial mass $m$ in an inertial frame, following a path $\gamma : [0,1] \to \mathcal{X}$. We assume constant velocity over the trajectory, i.e., $\|\gamma'(t)\|_2 = v_0$. From the law of conservation of momentum (no external forces), we have that the particle should shed mass according to the Tsiolkovsky rocket equation Tsiolkovsky (1954):

$$-m\frac{d\gamma'(t)}{dt} = (v_o(t) - \gamma'(t))\frac{dm(t)}{dt} \tag{37}$$

$$= v_e(t)\frac{dm(t)}{dt}, \tag{38}$$

where $v_o(t)$ stands for the output velocity of the expelled mass and $v_e(t)$ is the output velocity in the particle reference frame. Rearranging the above equation we obtain:

$$\frac{d\log(m(t))}{dt} v_e(t) = -\gamma''(t). \tag{39}$$

This equation shows that $v_e(t)$ and $\gamma''(t)$ are parallel vectors with different magnitudes. The above system of equations is thus equivalent to the following:

$$\frac{d\log(m(t))}{dt} \|v_e(t)\|_2 = -\|\gamma''(t)\|_2. \tag{40}$$

Assuming a constant relative norm for the output velocity $\|v_e(t)\|_2 = v_e$, and assuming no potential function in the system, we finally get the difference in log-energy of the particle between time $t = 0$ and $t = 1$ as

$$\log(E_1) - \log(E_0) = \log\left(\frac{m(1)}{m(0)}\right) = -\frac{1}{v_e}\int_0^1 \|\gamma''(s)\|_2\, ds. \tag{41}$$

The logarithm being a concave function, we obtain that minimizing the loss of energy along a path $\gamma$ amounts to minimizing the cumulative curvature along the path.

## J COMPUTATIONAL COMPLEXITY

Compared to CFM, the complexity of our method differs by two parameters: the number of snapshots $K + 1$ and the number of conditions $C$. Indeed, while CFM only considers the matching between two distributions, a single OT solution needs to be computed. In the MMFM case, $C \cdot K$ OT problems need to be solved. The complexity of our method is thus linear in the number of time points and in the number of conditions.

Optimal transport solvers typically show a quadratic dependence on the number of data points, although recent methods can marginally improve upon this bound Dvurechensky et al. (2018). When feasible, we pre-computed the OT couplings before training. However, when the datasets grow larger, we rely on mini-batching and solve OT couplings over mini-batches during training, following the approach of Tong et al. (2024). We note that our experiments all fall in the first scenario and OT-couplings could be computed before training. Given the successful applications using mini-batch OT approximations in Tong et al. (2024), we believe that MMFM with minibatch OT will demonstrate strong empirical performance as well.

We assessed the computational complexity of MMFM with respect to the number of time points, conditions, and total number of samples in a new benchmark experiment. We precomputed the solution to the OT problem. Our baseline scenario uses $K + 1 = 7$ time points, $C = 12$ conditions and $N = 50$ samples – following the setup of our synthetic experiment in Section 5.1, i.e. we generate N samples for each condition and each time point, resulting in $N \cdot (K + 1) \cdot C$ samples. We then varied these parameters independently to study their impact on running time. We kept all other hyperparameters and training details identical to the original experiment, executed on a GPU and results were averaged across three seeds. The computational time required for solving the Optimal Transport problem is indicated in parentheses beneath the run time. Results are shown in Table 9.

| No. Samples ($N$) | Time [$s$] |
| --- | --- |
| 50 | 27.45 |
| | (0.08) |
| 100 | 48.99 |
| | (0.10) |
| 500 | 232.86 |
| | (2.11) |
| 1000 | 461.31 |
| | (9.58) |

| No. Cond. ($C$) | Time [$s$] |
| --- | --- |
| 12 | 26.98 |
| | (0.09) |
| 24 | 49.62 |
| | (0.12) |
| 100 | 195.07 |
| | (0.72) |

| No. Timep. ($K+1$) | Time [$s$] |
| --- | --- |
| 7 | 26.12 |
| | (0.09) |
| 14 | 26.20 |
| | (0.19) |
| 28 | 27.73 |
| | (0.67) |

Table 9: Comparison of run time (and pre-computing the Optimal Transport solution, given in parentheses) with respect to different sample sizes, number of conditions, and time points. **Left**: We set $C = 12, K + 1 = 7$ and vary the sample size $N$. **Middle**: We set $N = 50, K + 1 = 7$ and vary the number of conditions $C$. **Right**: We set $N = 50, C = 12$ and vary the number of time points $K + 1$.

