# OpenReview forum: "Modeling Complex System Dynamics with Flow Matching Across Time and Conditions"
_ICLR.cc/2025/Conference — ICLR 2025 Spotlight_

### Official Review · Reviewer_Xe9p · 2024-11-02

**Soundness:** 2
**Presentation:** 1
**Contribution:** 2
**Rating:** 6
**Confidence:** 5

**Summary:**

The paper introduces a Multi-Marginal Flow Matching (MMFM) method to model the dynamics of complex systems from fragmented temporal snapshot data. MMFM constructs a flow using smooth spline-based interpolation across time points and experimental conditions, and regresses this flow with a neural network using a classifier-free guided flow matching framework. The paper demonstrates the effectiveness of MMFM on both synthetic and real-world datasets, including a single-cell genomics dataset with around a hundred chemical perturbations, showing the MMFM’s power in imputing data at missing time points.

**Strengths:**

-The use of smooth spline-based interpolation is a plausible approach for constructing the flow across different time points.
-The authors evaluate MMFM on a real-world dataset involving cellular responses to drugs, demonstrating the method's applicability to practical problems in biology and medicine.

**Weaknesses:**

Lack of Clear Problem Definition: The paper lacks a clear and concise problem definition. There is no centralized notation or formal problem statement, making it difficult to follow the methodology and understand the specific goals of the research.
Incomplete Methodological Details: The method section is incomplete and lacks crucial details. For example, the loss function used for training is not fully explained, and key components such as mean and variance formulas are missing. Additionally, the specific implementation of optimal transport is not described, leaving readers uncertain about how the method works in practice.
Unclear Experimental Setup: The experimental setup is poorly documented. The origin of the formulas used in the synthetic dataset is not explained, and the y-axis labels in Figure 2 are missing, making it difficult to interpret the results. For the real-world dataset on cellular responses to drugs, the criteria for evaluating performance are not clearly defined, making it hard to assess the effectiveness of MMFM.
Limited Baseline Comparisons: The performance comparison with baseline methods is inadequate. The paper mentions that MMFM outperforms existing methods but does not provide detailed comparisons or statistical significance tests, which are necessary to validate the claims.

**Questions:**

Why is the condition c multiplied by a scalar in Eq.(11) possible, as c is only the identifier of the condition?
What does the y-axis represent in Figure 2?
How is optimal transport specifically implemented in MMFM?
The MMFM loss function (7) and the CFM loss function (3) seem to be the same, what is the purpose of introducing the MMFM loss function?
From which literature do the formulas used in the synthetic dataset originate, What is the purpose of studying this synthetic dataset?
What is the exact definition of c_{\phi}?
What is the exact definition of \mu_{t}^{‘}(x) and \sigma_{t}^{‘}(x)?
What is the exact network architecture of \bar{v}_t(.)?

---

> ### Author Response · Authors · 2024-11-27
> **Official Comment by Authors**
>
> Dear Reviewer,
>
> Thank you for your detailed and thorough review.
>
> First, we would like to thank you for highlighting the well-motivated use of the smooth spline-based interpolation as well as the evaluation on real-world data.
>
> Second, we would like to address your feedback mentioned under “Weaknesses” and subsequently address each of the questions you raised explicitly. The revisions we brought to our work based on your suggestions already improved the paper’s technical rigor and clarity.
>
> ### Weaknesses
>
> **About the lack of a problem definition.**
>
> >`Lack of Clear Problem Definition: The paper lacks a clear and concise problem definition. There is no centralized notation or formal problem statement, making it difficult to follow the methodology and understand the specific goals of the research.`
>
> Despite being evoked in the introduction (“To leverage the rich and complex data now available and deliver better models, we propose framing the problem of learning cell population dynamics as modeling the transport of the probability distribution of cellular states across time and conditions. This approach allows us to capture changes over time and across conditions, while accommodating unpaired samples.”), we acknowledge that the problem definition was not clearly stated. We have now added an explicit problem statement in Section 3 and edited the background section to clarify our objectives.
>
> In general, we build upon the general motivation of the Flow Matching framework, which is to learn a vector field $u_t$ that generates a probability path $p_t(x)$ that satisfies the boundary conditions $p_0$ and $p_1$. Given an approximation of the target vector field $u_t(x)$, one can then easily sample from the distribution $p_1$ given samples from $p_0$ (or the other way around), as well as predict dynamics of individual samples over time.
>
> Our approach generalizes this framework by considering a collection of distributions $\{p_{t_k}(x\mid c):c\in[C],k\in [K+1]\}$ over $C$ different conditions and $K+1$ different sampling times. Given these observations, our goal is to learn an approximation of a target vector field $u_t(x\mid c)$ that generates a probability path $p_t(x\mid c)$ that satisfies the boundary conditions at $t_0,t_1,\dots,t_K$ for all conditions. Writing $\phi_t^c$ for the flow generated by $u_t(x\mid c)$, we want $[\phi_{t_k}^c]_{*} p_{t_0}(x\mid c) = p_{t_k}(x\mid c)$ for all conditions $c$ and sampling times $t_k$.
>
>
> **About missing methodological details.**
>
> >`The method section is incomplete and lacks crucial details. For example, the loss function used for training is not fully explained, and key components such as mean and variance formulas are missing.`
>
> Due to the page limit, we could unfortunately not explain each aspect in detail in the main text. However, we want to point out that the mean and variance formulas are presented. Indeed, Eq.9 presents the mean formula, while Eq.10 provides the variance formula. Additionally, the loss function is explained in the paragraph following Eq.12.
> To improve clarity, we have added additional details regarding the loss in Eq. 12, and have assigned a separate equation number to the variance function equation (which was part of the main text before).
>
> **About missing experiment setup details**
>
> >`The specific implementation of optimal transport is not described, leaving readers uncertain about how the method works in practice. [...]
> Unclear Experimental Setup: The experimental setup is poorly documented. The origin of the formulas used in the synthetic dataset is not explained, and the y-axis labels in Figure 2 are missing, making it difficult to interpret the results. For the real-world dataset on cellular responses to drugs, the criteria for evaluating performance are not clearly defined, making it hard to assess the effectiveness of MMFM.`
>
> As per the methods section, we acknowledge that some details were left out for meeting the page limit. Hence, we welcome the comments of the reviewer as they help us identify what important information is missing from the paper.
>
> We are pleased to report that we have improved the documentation of the experimental setup as per the reviewer’s suggestions. As described in detail in the replies to your specific questions below, we have edited Figure 2 for clarity, clarified the origin of the synthetic dataset, and have provided additional details regarding the experiments in Section 5.2 and Appendix D. Point-by-point replies and references to the updated manuscript to your questions are provided in the sections below.

---

> ### Author Response · Authors · 2024-11-27
> **Official Comment by Authors**
>
> ### Point-by-point replies to questions
>
> **About multiplying the condition identifier with a scalar**
> >`Why is the condition c multiplied by a scalar in Eq.(11) possible, as c is only the identifier of the condition?`
>
> The notation is based on the notation used in [1] (e.g., see Eq.11), which extends the classifier-free guidance [2] to Flow-Matching. We acknowledge this represents a slight abuse of notation. However, this notation is aimed at keeping the equations compact and clear to the reader.
> We first sample `b` from a Bernoulli distribution with probability `p_u` representing the probability of switching to the unconditional model. The value of `b` now indicates whether we select either the conditional model or the unconditional model:
> Given `b=0` we select the conditional model. The input to the neural network is then the condition label, `c`, which is used to retrieve a learnable embedding for the condition.
> If `b=1`, we select the unconditional model. The input to the neural network is `c_\varnothing`, which is used to retrieve a learnable embedding for the unconditional model.
> Overall, `c_\varnothing` can be seen as an additional condition that pulls all conditions together. This strategy is widely used in previous works, such as [1].
> We clarified this in Sec.3.2:
> `[$v_t(x, c; \theta)$] takes as input a condition index $c$ that is used to internally retrieve a learnable condition embedding (details about the architecture are provided in Appendix D) [...] The null conditioning $c_\varnothing$ is an additional condition index that retrieves a learnable embedding  corresponding to a condition-independent model. When the Bernoulli variable $b$ is 1, the model uses a conditional-independent model. Otherwise, the conditional model is used.`
>
> **About the definition of $c_{\varnothing}$**
>
> > “What is the exact definition of c_{\phi}?”`
>
> As described above, the null conditioning $c_\varnothing$ is an additional condition index that retrieves a learnable embedding corresponding to a condition-independent model. When the Bernoulli variable $b$ is 1, the model uses this condition-independent embedding instead of the embedding corresponding to the actual condition $c$, effectively making the prediction independent of any specific condition. We were inspired by the notation used in [1] and the flagship paper for classifier-free guidance [2]. We used this question to improve the paper's clarity by adding a more detailed explanation in the manuscript, see Sec.3.2.
> Excerpt from Section 3.2:
> `The null conditioning $c_\varnothing$ is an additional condition index that retrieves a learnable embedding  corresponding to a condition-independent model. When the Bernoulli variable $b$ is 1, the model uses a conditional-independent model. Otherwise, the conditional model is used.`
>
> **About the architecture of the vector field neural network**
>
> >`What is the exact network architecture of \bar{v}_t(.)?`
>
> Thank you for this question. We have now added a dedicated paragraph in Appendix D:
> `Given an observation $x$, a condition $c$, and a time $t$, we first compute a dimensional temporal embedding for $t$ using a sinusoidal timestep embedding (with a minimum frequency of 10000). We then retrieve a dimensional condition embedding from a library of learnable condition embeddings, using the index $c$. The input $x$ is transformed into a dimensional representation via a three-layer feedforward network using LeakyReLU activations. Subsequently, the embeddings of $x$, $t$, and $c$ are concatenated and processed through an additional three-layer neural network.`
> The latent dimensions used in the hyper-parameters search are given in Table 5 of the same Appendix. We refer to Appendix D for the architecture of the network in Section 3.2, right after Equation (11):
> `We then integrate the condition as an additional input to the neural network, modifying it to $v_t(x, c; \theta)$. This neural network takes as input a condition index $c$ that is used to internally retrieve a learnable condition embedding (details about the architecture are provided in Appendix D.`

---

> ### Author Response · Authors · 2024-11-27
> **Official Comment by Authors**
>
> **About the definition of $\mu$ and $\sigma$**
>
> > `What is the exact definition of \mu_{t}^{‘}(x) and \sigma_{t}^{‘}(x)?`
>
> Thank you for highlighting the importance of the derivatives of the mean and variance functions. Since the derivatives are essential for computing the conditional vector field in Eq.6, we want to clarify how both functions and their corresponding derivatives were computed.
>
> **Mean Function**
> The definition of `\mu_t(z)` can be found in Eq.9 and represents the natural cubic spline-based mean-function of the flow. Hence, the mean function $\mu_t(z)$ is a piecewise third-order polynomial function with $\mathcal{C}^2$ continuity, meaning its values and first and second derivatives are continuous at interval boundaries. For each interval $[t_k, t_{k+1}]$, the spline can be expressed as:
> $$\mu(t,z) = a_k(t - t_k)^3 + b_k(t - t_k)^2 + c_k(t - t_k) + d_k,$$
> where $a_i$, $b_i$, $c_i$, and $d_i$ are coefficients specific to that interval. Therefore, the first derivative is:
> $$\mu^{'}(t,z) = 3a_k(t - t_k)^2 + 2b_k(t - t_k) + c_k.$$
> The spline and its derivative can be efficiently computed by solving a tridiagonal system of linear equations.
>
> **Variance Function**
> For the variance function `\sigma_t(z)`, we considered two options:
> - Constant values and therefore $\sigma_t^{’}(z)=0$
> - If $\sigma_t(z)$ is as defined in Eq.10, then we have $\sigma_t^{'}(z) = 4 \cdot \frac{t_k + t_{k+1} - 2 t}{(t_{k+1} - t_k)^2$.
>
> Both of these formulas are also efficient to compute.
> We have included these derivations in Appendix D and refer to it right after Equation 10 in the main text.
>
> **About the implementation of the OT map**
>
> > `How is optimal transport specifically implemented in MMFM?`
>
> As described in the `Optimal Transport` paragraph of Section 3.1., we set `q(z)` to be the multi-marginal optimal transport map between distributions $$\{ p_{t_k}(\cdot \mid c): t\in [K+1]  \}$$. As shown in Appendix B, we assume the cost function decomposes as follows:
> $$c(x_{t_0},x_{t_1},\dots , x_{t_K}) = \prod_{0}^{K-1} c(x_{t_k},x_{t_{k+1}}) $$
> and that each pairwise cost function is the squared Euclidean distance cost $c(x_{t_k}, x_{t_{k=1}}) =\mid x_{t_k} −  x_{t_{k=1}} \mid^2$,
> we thus have that $q(z)$ is given by
>
> $$q\^{c}(z\^{c}) = \pi\_{c}\^{\*,(t\_{0},t\_{1})} (z\^{c}\_0,z\^{c}\_1) \pi\_{c}^{\*,(t\_{1},t\_{2})} (z\^{c}\_{2} \mid z\^{c}\_{1}) \dots \pi^{\*,(t\_{K-1},t\_{K})}\_c (z^c\_{K-1} \mid z^c\_{K})$$
>
> where  $\pi^{*,(t\_k,t\_{k+1})}\_c(z^c\_{k} \mid z^c\_{k+1})$ is the 2-Wasserstein optimal transport map between $p\_{t\_k}(\cdot \mid c)$ and $p\_{t\_{k+1}}(\cdot \mid c)$.
>
> That is, $\pi^{*,(t\_k,t\_{k+1})}\_c(z^c\_{k} \mid z^c\_{k+1}) = \inf\_{\pi \in \Pi} \int\_{\mathbb{R}^d \times \mathbb{R}^d} c(x\_{t\_k},x\_{t\_{k+1}})^2 d\pi(x\_{t\_k},x\_{t\_{k+1}}),$
>
> Practically, we solve the above problem for each $(t_k,t_{k+1)}$ and $c$ using the `ot.emd2` function provided in the Python package POT (v0.9.4). Please note that we compute the exact solution in our experiments and do not use Sinkhorn algorithm-based approximations.
>
> We acknowledge that these important details should be included in the main text and have thus expanded our discussion in the Optimal Transport paragraph of Section 3.1. Appendix D includes details about the numerical OT implementation in a dedicated paragraph.
>
>
> Excerpt from Section 3.1.:
> `Optimal Transport [...] [we] propose to sample data points across marginals using a joint distribution, computed as the solution of a multi-marginal optimal transport (MMOT) problem: ` $q(z)=\pi^*(x_0,...,x_K)$. `Notably, when the cost structure is pairwise additive between each pair of marginals, the MMOT problem reduces to a set of ` $K$ `independent OT problems, such that` $\pi^*(x\_0,...,x\_K) =  \frac{\prod\_{k=1}^{K-1}\pi^*\_k(x\_k, x\_{k+1})}{\prod\_{k=2}^{K-1}p\_{t_k}(x\_k)} $, where $\pi^*\_k(x\_k, x\_{k+1})$ `is the solution of the OT problem between` $p\_{t\_k}$ and $p\_{t\_{k+1}}$. `(The proof is provided in Appendix B). Throughout this work, we assume such pairwise additive cost structure. We pre-compute the solutions of the OT problems on the training data using the squared Euclidean distance as a cost function and use the multi-marginal optimal transport coupling for` $q$ `in Equation 7. When the dataset is too large, OT can be approximated using mini-batches, although this was not necessary in our experiments.`
>
>
> We also included more details in Section 3.2. for the condition-dependent coupling.

---

> > ### Author Response · Authors · 2024-11-27
> > **Official Comment by Authors**
> >
> > Dear Reviewer,
> >
> > Unfortunately, we are currently experiencing issues with the proper rendering of LaTeX formulas in our responses. We are actively working to resolve this problem and hope to resolve it shortly. We apologize for the inconvenience this may cause.

---

> ### Author Response · Authors · 2024-11-27
> **Official Comment by Authors**
>
> >`The MMFM loss function (7) and the CFM loss function (3) seem to be the same, what is the purpose of introducing the MMFM loss function?`
>
> We appreciate your attention to detail when reviewing our manuscript and realizing that Equation 7 and Equation 3 have the same mathematical structure. We present this objective another time in Equation 7 for two important reasons:
> 1) The underlying random variable $z$ now follows a joint distribution over variables at multiple time steps, $z \coloneq(x_0, \ldots, x_K) \in \mathcal{Z} = \mathcal{X}^{K+1}$. This results in different conditional and marginal density paths $u_t(x\mid z)$ and $u_t(x)$. We believe that the restatement enhances the clarity by providing readers with the complete context of the optimization objective and its modified components.
> 2) Restating the equation explicitly establishes the MMFM objective's dependency on the new density and the new path, enabling precise references in Proposition 1 and throughout the manuscript. This approach eliminates the need for additional notation to distinguish between the objective for the classic Flow Matching and its density path and the MMFM spline-based version.
> We clarified this after Equation 7:
> `Note that this objective has the same form as Equation \ref{eq:cfm}, but the density $q(z)$ and the probability density path is defined differently (over multiple time steps), which impact the target conditional vector field $u_t$.`
>
> **About Figure 2**
>
> >`What does the y-axis represent in Figure 2?`
>
> Thank you for your detailed study of the figures which we used to improve the manuscript. The plots show a phase diagram of the ground truth and learned vector fields. Since we study and plot 2-dimensional vectors `x` in the synthetic experiment, we relabelled the x- and y-axis to `x^0` and `x^1` in the updated manuscript to define both dimensions and to prevent confusion with the subscript indicating the time, `x_t`.
>
> **About the sources of the synthetic experiment**
>
> > `From which literature do the formulas used in the synthetic dataset originate. What is the purpose of studying this synthetic dataset?`
>
> We constructed this synthetic example as an extension of Figure 1 in [3]. We modified it to have a continuous vector field and multiple conditions influencing the trajectories. We presented this example to provide an intuitive understanding for the reader of the scenarios in which MMFM can perform well and show improvements over competitor methods.
> Such trajectory patterns are prevalent in real-world datasets, including:
> Weather data exhibiting seasonality effects
> Urban traffic flow patterns influenced by rush hour dynamics
> Resistor-capacitor circuits displaying alternating voltage profiles
> These systems often face challenges of missing data due to high measurement costs, data loss or data corruption. Consequently, imputation techniques are crucial for reconstructing a comprehensive view of the underlying dynamics. Our synthetic experiment mimics these real-world systems, effectively demonstrating MMFM's ability to address complex trajectory reconstruction and data imputation challenges in multi-conditional, continuous vector field scenarios.
> Again, thank you for your valuable feedback, which has significantly enhanced our manuscript. We hope to have addressed all the points raised in your review. We are confident that the improvements made have strengthened the overall quality and clarity of our work. Should any additional questions or concerns arise, we remain open to further discussion.
>
>
> References:
> [1] Zheng, Qinqing, et al. "Guided flows for generative modeling and decision making." arXiv preprint arXiv:2311.13443 (2023).
> [2] Ho, Jonathan, and Tim Salimans. "Classifier-free diffusion guidance." arXiv preprint arXiv:2207.12598 (2022).
> [3] Chewi, Sinho, et al. "Fast and smooth interpolation on wasserstein space." International Conference on Artificial Intelligence and Statistics. PMLR, 2021.

---

> ### Author Response · Authors · 2024-12-01
> **Official Comment by Authors**
>
> Dear reviewer,
>
> as the end of the rebuttal period is approaching, we would be happy to answer any remaining or additional questions.
> We would like to emphasize that we have carefully incorporated your suggestions to enhance the clarity of our manuscript and addressed your questions and comments in our rebuttal answers.
> We appreciate your time and effort in reviewing our work. If you find our rebuttal and additions to the manuscript satisfactory, we would be grateful if you could consider increasing your score.
> Should you have any further points or questions, we are more than willing to engage in additional discussion.

---

> ### Author Response · Authors · 2024-12-03
> **Last hours of discussion period**
>
> Being in the last hours of the discussion period, we are still eager to answer any remaining concern the reviewer may have. We also fixed the latex rendering issues in our responses above.
>
> We would like to emphasize that we have carefully incorporated your suggestions to enhance the clarity of our manuscript and addressed your questions and comments in our rebuttal answers. We appreciate your time and effort in reviewing our work. If you find our rebuttal and additions to the manuscript satisfactory, we would be grateful if you could consider increasing your score. Should you have any further points or questions, we are more than willing to engage in additional discussion.

---

### Official Review · Reviewer_FqP7 · 2024-11-03

**Soundness:** 3
**Presentation:** 3
**Contribution:** 3
**Rating:** 8
**Confidence:** 3

**Summary:**

This paper introduces Multi-Marginal Flow Matching (MMFM), a method for modeling the dynamics of biological single-cell time series data. The proposed approach extends the Flow Matching (FM) framework by incorporating multiple time points and conditions, thereby learning the transport of probability distributions for cellular states across these different scenarios. MMFM leverages techniques like smooth spline-based interpolation and neural networks with classifier-free guidance to model changes over time and conditions. Through experiments on synthetic and single-cell RNA-seq datasets, the authors show MMFM's superior performance in imputing missing time points compared to other baseline methods.

**Strengths:**

- The motivation for using MMFM to model cell population dynamics across multiple steps and conditions is well-founded, as it directly addresses challenges inherent in biological experiments, such as fragmented and incomplete data. The multi-step transport approach is impressive in efficiently capturing the system's temporal and conditional complexities.

- The method extends (Conditional) Flow Matching (FM) to the *Multi-Marginal** setting, which is a natural and well-justified progression for addressing biological systems that require modeling across multiple time points. This approach enables the framework to leverage inter-dependencies across multiple conditions, significantly enhancing generalizability.

 - The use of **optimal transport** and **Schrödinger Bridge** for improving the integration of multiple marginals is insightful, resulting in improved model robustness. Compared to **piece-wise linear interpolation**, the introduction of cubic splines for smoother interpolation significantly boosts the performance and efficiency of the model, which is well demonstrated in their experiments.

**Weaknesses:**

- The title, **"Modeling Complex System Dynamics,"** might be an over-claim relative to the paper's focus, which is primarily on **biological single-cell dynamics**. Typically, "Complex System Dynamics" also refers to broader scenarios, such as physical phenomena in fluid dynamics, climate systems, or even economic time series. Since the proposed method appears promising and generalizable, demonstrating its applicability to tasks in broader domains would further strengthen this claim.

- In the Multi-Marginal Conditional Flow Matching (MMCF), the authors pre-compute the solution for the multi-marginal optimal transport, which can be time-consuming. While they mention that **mini-batches** can be used to handle large datasets, further discussion on the **scalability** of the approach would be valuable. In particular, it would help if the authors could expand on the tradeoff between the **performance** and **efficiency** when comparing **L-MMFM** and **MMCF**, as OT solvers tend to become computationally expensive with increased dimensionality.

- Although MMFM is demonstrated effectively on biological single-cell data, there is no extensive discussion about its generalization capabilities in handling diverse temporal datasets beyond biology. Considering the potential to model other types of time series (e.g., physical or environmental data), the paper would benefit from a discussion about such possible applications and any inherent limitations.

**Questions:**

- Some generative models designed for imputation task can be also adapted for extrapolation/forecasting, such as CSDI[1]. Can the MMFM or CSD method be extended to handle **extrapolation** for sequences beyond the given time points?

- The current model appears to rely on temporal snapshots at specific intervals. Can **MMFM** handle **non-uniformly sampled** time points? If so, how is this addressed within the interpolation or flow matching framework?

[1]: Tashiro, Yusuke, et al. "Csdi: Conditional score-based diffusion models for probabilistic time series imputation." Advances in Neural Information Processing Systems 34 (2021): 24804-24816.

---

> ### Author Response · Authors · 2024-11-27
> **Official Comment by Authors**
>
> Dear Reviewer,
>
> Thank you for your thoughtful and detailed review. We particularly appreciate your recognition of our work's strong motivation and justified extension of CFM, especially noting how our framework leverages inter-dependencies across multiple conditions. Your positive feedback on our empirical results is also appreciated.
>
> You can find our detailed responses to your questions and concerns below. We have incorporated all suggested revisions into the manuscript, with modifications highlighted in blue throughout the main text and appendix. In particular, we have now included experiments on an additional dataset of air pollution measurements that further shows the advantages of our approach.
>
> Addressing Your Questions:
>
> **Title Scope**
>
> > `The title, "Modeling Complex System Dynamics," might be an over-claim relative to the paper's focus, which is primarily on biological single-cell dynamics. [...] Since the proposed method appears promising and generalizable, demonstrating its applicability to tasks in broader domains would further strengthen this claim.`
>
> We acknowledge your comment regarding the current title "Modeling Complex System Dynamics" being too broad given our results were hitherto focused on synthetic and single-cell dynamics. First we would like to mention, the method is not particularly designed for single-cell data and therefore can be used to model dynamics in any field, such as finance, physics or chemistry. Second, we added another experiment to show the generalizibility of our method. We applied MMFM to the “Beijing Multi-Site Air Quality” [3] dataset, which contains hourly PM2.5 pollution measurements from 12 monitoring sites between January 2015 and February 2017. The experiment addresses sensor network challenges where measurements require imputation due to data loss, corruption or high costs. We selected training data points with varying temporal gaps of 1-7 months to assess the model's capability in handling non-uniform sampling. Both MMFM variants demonstrated superior performance in interpolating sparse station measurements, with L-MMFM achieving the highest Maximum Mean Discrepancy scores, followed by cubic spline-based MMFM. Comprehensive experimental details and evaluations are provided in Section 5.3 and Appendix G of our manuscript.
>
> **Extrapolation Capabilities**
>
> >`Some generative models designed for imputation task can be also adapted for extrapolation/forecasting, such as CSDI. Can the MMFM or CSD method be extended to handle extrapolation for sequences beyond the given time points?`
>
> Regarding the potential for extrapolation, we would like to highlight two distinct scenarios for extrapolating trajectories at time $t$ for condition $c$:
> - There exist other conditions $c’$ observed at time $t$.
> - No conditions have been observed at time $t$ and later than $t$.
>
> The first scenario is already covered in our manuscript, e.g., the single-cell experiment of Section 5.2. In Table 3, the results for the 72h experiment represent the performance at predicting the distributions of a condition at the last observed time point given observations at 0, 24, and 48 hours, as well as the observations for *other conditions* at 0, 24, 48, and 72 hours time points. Our results show that our model performs well at this task and outperforms other approaches. Additionally, we provide a similar extrapolation example using simulated data in Appendix F.
>
> The second scenario is more complicated and requires strong assumptions about the dynamics of the process under study. Indeed, this corresponds to forecasting beyond a horizon that has never been observed before and cannot thus be learnt from data alone. To draw an analogy from the clinical domain, it amounts to predicting what would happen to a patient 1 year after giving a new therapy, given that the longest follow-up available in the data is 6 months. There is no data to support that prediction, and strong inductive biases mirroring prior expert knowledge must be used instead. Now, our method learns a vector field that can be evaluated beyond the most prolonged time horizon seen in the data. Hence, nothing forbids computationally forecasting beyond that time point. However, the extrapolation performance would heavily depend on the validity of the inductive bias of our method, such as the natural cubic spline mean function.

---

> ### Author Response · Authors · 2024-11-27
> **Official Comment by Authors**
>
> **Non-uniform Sampling**
>
> > `The current model appears to rely on temporal snapshots at specific intervals. Can MMFM handle non-uniformly sampled time points? If so, how is this addressed within the interpolation or flow matching framework?`
>
> We appreciate the opportunity to clarify a potential misunderstanding and highlight one of our method’s benefits: MMFM is indeed capable of handling non-uniformly sampled time points. That is, we don’t assume any regularity in the sampling times of the snapshots. As Table 1 shows, our synthetic experiments contain measurements at irregular time steps (with sampling intervals 0.1, 0.05, 0.15, or 0.2). Moreover, in our newly added experiment on the “Beijing Multi-Site Air Quality” [3] data set, we leverage non-uniformly sampled time points. In this case, we observe measurements with time intervals between 1 and 7 months on a 26 month long horizon. However, we assume multiple observations per snapshot (hence multiple observations simultaneously). When this is not possible, a binning strategy could be used to create such snapshots from the data.
>
> We added an additional sentence to Sec 3.1 to improve the manuscript's clarity.
>
> *Excerpt from Section 3.1*:
> `Note that we do not impose any constraints on the spacing between consecutive time points. This flexibility accommodates scenarios where data collection occurs at irregular intervals, which is often the case in real-world applications.`

---

> ### Author Response · Authors · 2024-11-27
>
> **Scalability**
>
> >`The authors pre-compute the solution for the multi-marginal optimal transport, which can be time-consuming. While they mention that mini-batches can be used to handle large datasets, further discussion on the scalability of the approach would be valuable. In particular, it would help if the authors could expand on the tradeoff between the performance and efficiency when comparing L-MMFM and MMCF, as OT solvers tend to become computationally expensive with increased dimensionality.`
>
> We agree that the computational efficiency of OT-based methods is an important aspect that deserves further discussion. We have added a full section in the Appendix (Appendix I) to discuss the computational complexity and provide runtime measurements.
>
> Regarding the trade-off between CFM and MMFM, two parameters come into play in the computational complexity: the number of snapshots $K+1$ and the number of conditions $C$. Indeed, while CFM only considers the matching between two distributions, a single OT solution needs to be computed. In the MMFM case, $C\cdot K$ OT problems need to be solved. The complexity of our method is thus linear in the number of time points and in the number of conditions.
>
> We assessed the computational complexity of MMFM with respect to the number of time points, conditions, and total number of samples in a new benchmark experiment. We precomputed the solution to the OT problem. Our baseline scenario uses $K+1=7$ time points, $C=12$ conditions and $N=50$ samples – following the setup of our synthetic experiment in Section 5.1, i.e. we generate N samples for each condition and each time point, resulting in $N \cdot (K+1) \cdot C$ samples. We then varied these parameters independently to study their impact on runtime. We kept all other hyperparameters and training details identical to the original experiment, executed the benchmark on a GPU and results were averaged across three seeds. The computational time required for solving the Optimal Transport problem is indicated in parentheses beneath the run time.
>
> | **No\. Samples \($N$\)** | **Time \[$s$\]** |
> |-------------------------------------:|-----------------------------:|
> | 50                               	| 27\.45                   	|
> |                                  	| \(0\.08\)                	|
> | 100                              	| 48\.99                   	|
> |                                  	| \(0\.10\)                	|
> | 500                              	| 232\.86                  	|
> |                                  	| \(2\.11\)                	|
> | 1000                             	| 461\.31                  	|
> |                                  	| \(9\.58\)                	|
>
>
> | No\. Cond\. \($C$\) 	| Time \[$s$\] 	|
> |-------------------------------------:|-----------------------------:|
> | 12                               	| 26\.98                   	|
> |                                  	| \(0\.09\)                	|
> | 24                               	| 49\.62                   	|
> |                                  	| \(0\.12\)                	|
> | 100                              	| 195\.07                  	|
> |                                  	| \(0\.72\)                	|
>
> | No\. Timep\. \($K\+1$\)  | Time \[$s$\] 	|
> |-------------------------------------:|-----------------------------:|
> | 7                                	| 26\.12                   	|
> |                                  	| \(0\.09\)                	|
> | 14                               	| 26\.20                   	|
> |                                  	| \(0\.19\)                	|
> | 28                               	| 27\.73                   	|
> |                                  	| \(0\.67\)                	|
>
> Regarding the number of samples. Optimal transport solvers typically show a quadratic dependence on the number of data points, although recent methods can marginally improve upon this bound [2]. When feasible, we pre-compute the OT couplings before training. However, when the datasets grow larger, we rely on mini-batching and solve OT couplings over mini-batches during training, following the approach of [1]. We note that our experiments all fall in the first scenario and OT-couplings could be computed before training. Given the successful applications using minibatch OT approximations in [1], we believe that MMFM with minibatch OT will demonstrate strong empirical performance as well.
>
> Once again, we appreciate your constructive feedback, which has substantially improved our manuscript. We hope that our revisions address all points raised and that our updated manuscript meets the requirements for publication now. We remain open to further discussion on these or any other emerging points.

---

> ### Author Response · Authors · 2024-11-27
> **Official Comment by Authors**
>
> References:
> [1] Tong, Alexander et al. “Improving and generalizing flow-based generative models with minibatch optimal transport.” Trans. Mach. Learn. Res. 2024 (2023): n. pag.
> [2] Dvurechensky, Pavel, Alexander Gasnikov, and Alexey Kroshnin. "Computational optimal transport: Complexity by accelerated gradient descent is better than by Sinkhorn’s algorithm." International conference on machine learning. PMLR, 2018.
> [3] Chen, S. (2017). Beijing Multi-Site Air Quality [Dataset]. UCI Machine Learning Repository. https://doi.org/10.24432/C5RK5G.

---

> ### Author Response · Authors · 2024-12-01
> **Official Comment by Authors**
>
> Dear reviewer,
>
> as the end of the rebuttal period is approaching, we would be happy to answer any remaining or additional questions!
>
> We would like to underline that we used your suggestions to improve the clarity and quality of the manuscript, provided details on all questions you raised in the rebuttal, as well as provided a new experiment to show the generalizability of our method to strengthen the claim in our manuscript's title.
> We thank you again for your time and effort!

---

> > ### Comment · Reviewer_FqP7 · 2024-12-03
> >
> > I appreciate the detailed rebuttal of the authors. It resolved some of my concerns and I will keep my socre to support the paper.

---

### Official Review · Reviewer_19bk · 2024-11-03

**Soundness:** 3
**Presentation:** 4
**Contribution:** 4
**Rating:** 8
**Confidence:** 3

**Summary:**

This paper aims to model the dynamics of complex real-world systems across time and conditions using flow matching. Current methods, such as Flow Matching, remain limited by the inability to effectively capture information across multiple time points and experimental conditions. This paper aims to alleviate these limitations with a novel method they refer to as  Multi-Marginal Flow Matching, which leverages the Flow Matching framework and optimal transport theory to train a neural network to regress onto the flow. The authors evaluate the method’s effectiveness on synthetic and real-world data, demonstrating superior performance to existing baselines, measured by qualitative and quantitative metrics.

**Strengths:**

This paper introduces a novel method which expands on previous methodology in a parsimonious manner.
The work is relevant to the research community, addressing the problem of modeling complex system dynamics.
The approach is simple and intuitive, with sound theoretical backing. The empirical evidence they provide is strong but limited.
The experiments are well-designed and include sufficient baselines. The authors appear to know the literature very well and do a good job providing citations throughout.
The paper is well-written and organized and easy to follow. I particularly appreciate how the authors clearly provide the minimally sufficient information in the main body of the paper and detail the appendix to complement.

**Weaknesses:**

The authors rigorously compare their method to multiple existing methods. However, they only do so on one real-world rna-seq dataset and one synthetic dataset. In the introduction of the paper, the authors give a multitude of real-world settings where their method may be considered useful. While the authors clearly motivate their method, I am concerned by the rigor with which they evaluate their method empirically. Are there additional datasets that could further demonstrate the effectiveness of the method, such as some of the genomics datasets used in Tong et al. (2024)? If so, inclusion would greatly increase this paper's contribution. The authors address this weakness in their rebuttal, providing additional experiments supporting their method's usefulness in practice.

**Questions:**

In the proof of proposition 1, the final (*), I am confused how Fubini’s theorem applies here to switch the order of the inner product and integral.
I may have a misunderstanding: is it necessary for conditions {c} to be given by the data. If they are missing, how may one proceed? In many real word systems, for example, conditions or categories may be latent and unobserved.

---

> ### Author Response · Authors · 2024-11-27
> **Official Comment by Authors**
>
> Dear Reviewer,
>
> Thank you for your thoughtful and constructive feedback on our submission. We greatly appreciate your time and effort in reviewing our work - especially highlighting the novelty of our method, its relevance to the research community, the simplicity and intuitiveness of our approach, the theoretical foundations, our literature knowledge, and the clarity of our presentation. We agree that trajectory inference is an important research topic, especially for single-cell biology research.
>
> You can find our detailed responses to your questions and concerns below. We have incorporated all suggested revisions into the manuscript, with modifications highlighted in blue throughout the main text and appendix. In particular, we have now included experiments on an additional dataset of air pollution measurements that further shows the advantages of our approach.

---

> ### Author Response · Authors · 2024-11-27
> **Official comment by Authors**
>
> Addressing Your Questions:
>
> **Necessity of Conditions $c$**
>
> > `I may have a misunderstanding: is it necessary for conditions {c} to be given by the data. If they are missing, how may one proceed? In many real word systems, for example, conditions or categories may be latent and unobserved.`
>
> You raised an important question about the necessity of conditions $c$ being given by the data and how one might proceed if they are missing, particularly in real-world systems where conditions or categories may be unobserved.
>
> Our work indeed assumes that the conditions are given by data. That is, each sample is annotated with a condition. This assumption is typically satisfied in single-cell perturbation experiments or climate data (as shown in our experiments). Nevertheless, we acknowledge that this requirement may limit the applicability of our method in scenarios where such conditions are unavailable or unobserved. This is an interesting direction that could lead to future research. We added a clarification to the manuscript in Sec.3.2.
>
> Excerpt from Section 3.2 :
> `We now extend our framework to consider cases where each observation $x_k^c$ from time point $t_k$ is associated with a condition $c \in \{1, \ldots, C\}$. \textcolor{blue}{We assume the condition information is provided for each observation}.`
>
> **Proof of Proposition 1**
>
> > `In the proof of proposition 1, the final (*), I am confused how Fubini’s theorem applies here to switch the order of the inner product and integral.`
>
> Thank you for raising this point about Proposition 1. We have clarified the steps in the manuscript. Swapping the order between the inner product and integral is justified because the inner product is a linear operator in each of its arguments (like the expectation), not due to Fubini's theorem. Fubini's theorem is only used to transform the sequential integration into a double integral. We've added these intermediate steps and their justifications to clarify the proof (see Appendix A of the updated manuscript).
>
> **Data**
>
> > `Are there additional datasets that could further demonstrate the effectiveness of the method, such as some of the genomics datasets used in Tong et al. (2024)?`
>
> We first comment on the usability of the dataset mentioned in your review and presented in Tong et al. (2024). Tong et al. (2024) present applications of OT-CFM to three genomic datasets: CITE-seq and Multiome datasets from the OpenProblem NeurIPS competition [1] as well as the Embryoid body data from [2]. Those three data sets have distinct time points (more than three) but only one unique condition; therefore, the datasets are unsuitable to show the effectiveness of MMFM in leveraging information across conditions.
>
> To demonstrate the generalizability and effectiveness of our method, we ran an additional analysis and applied MMFM to the "Beijing Multi-Site Air Quality" data set [3]. This dataset comprises hourly air pollution data from 12 air-quality monitoring sites across Beijing. We focused on the PM2.5 pollution data measurements between January 2015 and February 2017. The air pollution data was grouped by month, showing a seasonality effect. Out of the 26 months, we selected multiple as training data, such that the difference between the timepoints varies between 1 and 7 months (highlighting the model’s capabilities of dealing with non-uniformly sampled data). We evaluated our method on unseen test months.
> This experiment addresses a common challenge in sensor networks, where some measurements are lost or corrupted and must be imputed using data from other sensors.
> The results indicate that both MMFM versions (cubic spline and linear paths) show a strong performance compared to FSI, CFM, and PCFM when it comes to interpolating the measurements of stations with only a few measurements. L-MMFM models shows the best performance across all evaluated methods, as measured by Maximum Mean Discpreancy (MMD), and the cubic spline-based MMFM becomes 2nd across all conditions. These findings have been incorporated into the manuscript (Section 5.3 and Appendix G), which provides comprehensive details and evaluations of the experiments.
>
>
> Once again, we appreciate your constructive feedback, which has substantially improved our manuscript. We hope that our revisions address all points raised and that our updated manuscript meets the requirements for publication now.
> We have implemented the reviewer's recommendation to perform another analysis on an additional real-world dataset underlying the generalizability of the framework. Furthermore, we have clarified the conditions necessary for MMFM and provided more details on one of our proofs.
> We remain open to further discussion on these or any other emerging points. Should our rebuttal adequately address their concerns, we would appreciate a reevaluation of our work, potentially leading to an improved score.

---

> > ### Comment · Reviewer_19bk · 2024-12-03
> >
> > I appreciate your detailed rebuttal and have raised my score accordingly. In particular, thank you for clarifying why you do not use the datasets used in Tong et al. (2024) and for including additional experiments showcasing MMFM's generalization capabilities.

---

> ### Author Response · Authors · 2024-11-27
> **Official Comment by Authors**
>
> References:
> [1] Lance, Christopher, et al. "Multimodal single cell data integration challenge: results and lessons learned." BioRxiv (2022): 2022-04.
> [2] Moon, Kevin R., et al. "Visualizing structure and transitions in high-dimensional biological data." Nature biotechnology 37.12 (2019): 1482-1492.
> [3] Chen, S. (2017). Beijing Multi-Site Air Quality [Dataset]. UCI Machine Learning Repository. https://doi.org/10.24432/C5RK5G.

---

> ### Author Response · Authors · 2024-12-01
> **Official Comment by Authors**
>
> Dear Reviewer,
>
> As the end of the rebuttal period approaches, we welcome any remaining or additional questions you may have. We would like to emphasize that we have followed your suggestions, adding clarifications to enhance the manuscript's clarity and providing new experimental evidence to demonstrate the generalizability of our method.
>
> We sincerely appreciate your time and effort in reviewing our work. If you find our rebuttal and additions to the manuscript satisfactory, we would be grateful if you could consider increasing your score.

---

### Meta-Review · Area_Chair_5Jy9 · 2024-12-19

**Metareview:**

The paper introduces Multi-Marginal Flow Matching, that generalizes Flow Matching to learn system dynamics from populations measured at multiple time points and under various conditions. The proposed framework allows for interpolation and extrapolation to unobserved
time points and conditions. The approach is demonstrated on both simulated and real data regarding cellular trajectory prediction and air pollutants concentration prediction.

The paper well-motivated and tackles important limitations of traditional flow matching algorithms. The approach is sound and achieves convincing results.

The authors have significantly improved their manuscript to address key concerns, incorporating valuable clarifying comments and including a new dataset evaluation.

**Additional Comments On Reviewer Discussion:**

The authors have done a great job at addressing the reviewer points. They have provided additional explanation e.g. regarding handling non-uniformity, applicability of alternative methods, relevance certain datasets. They  have clarified the manuscript e.g. to clearly formulate the problem and they have included evaluation on a dataset pertaining to air quality prediction.

The reviewers have raised or maintained their already positive scores and the AC agrees that this work is worthy of acceptance at ICLR.

---

### Decision · Program_Chairs · 2025-01-22

Accept (Spotlight)